# Scout Before You Attend: Sketch-and-Walk Sparse Attention for Efficient LLM Inference

Hoang Anh Duy Le [1 2]  Sahil Joshi [1]  Zeyu Yang [1]  Zhaozhuo Xu [2]  Anshumali Shrivastava [1]

## Abstract

Self-attention dominates the computational and memory cost of long-context LLM inference across both prefill and decode phases. To address this challenge, we introduce **Sketch&Walk** Attention, a training-free sparse attention method that determines sparsity with lightweight sketches and deterministic walk. Sketch&Walk applies Hadamard sketching to get inexpensive approximations of attention scores, then aggregates these estimates across layers via a walk mechanism that captures attention influence beyond direct interactions between tokens. The accumulated walk scores are used to select top-$k$ attention blocks, enabling dynamic sparsity with a single training-free algorithm that applies uniformly to both the prefill and decode phases, together with custom sparse attention kernels. Across a wide range of models and tasks, Sketch&Walk maintains near-lossless accuracy at 20% attention density and can slightly outperform dense attention in some settings, while achieving up to $4.7\times$ end-to-end attention speedup over FlashAttention-2.

## 1. Introduction

Large Language Models (LLMs) have demonstrated remarkable capabilities, yet their deployment at scale remains hindered by the quadratic cost of self-attention (Vaswani et al., 2017), which becomes prohibitive as context lengths grow to hundreds of thousands of tokens (Pope et al., 2023; Fu, 2024). Numerous efficient inference algorithms have been proposed to address this bottleneck (Yuan et al., 2024), including sparse attention methods that compute only a subset of query-key interactions (Jiang et al., 2024; Tang et al., 2024; Lai et al., 2025; Yan et al., 2025). Despite their effectiveness, most existing sparse attention methods rely on

---

[1]Department of Computer Science, Rice University [2]Workato. Correspondence to: Zhaozhuo Xu <zhaozhuo.xu@workato.com>.

*Proceedings of the 43rd International Conference on Machine Learning*, Seoul, South Korea. PMLR 306, 2026. Copyright 2026 by the author(s).

a common design choice: they select key blocks based on *one-hop* attention scores computed independently at each layer. This one-hop selection inherently misses *multi-hop* token connections (where a query block $i$ depends on block $k$ through intermediate blocks $j$) that emerge only through repeated attention composition in deep transformers (Abnar & Zuidema, 2020; Cai et al., 2024).

**One-hop-per-layer selection misses multi-hop token dependencies.** In transformers, information propagates through layers via repeated attention composition. As a result, the dependence of a token at position $i$ on another token $k$ may arise through a sequence of intermediate tokens, rather than through a single direct interaction. However, attention scores are inner products, a non-metric similarity that violates transitivity: if block $i$ attends strongly to $j$, and $j$ to $k$, this does *not* imply $i$ directly attends strongly to $k$. Yet, existing sparse attention methods compute one-hop scores at each layer and rely on cross-layer stacking to implicitly capture multi-hop dependencies. However, this might *fail*: if an intermediate token $j$ is not selected at layer $\ell$, any multi-hop dependency of the form $i \to \ldots \to j \to k$ is broken, and subsequent layers cannot recover it. Moreover, this limitation *cannot be resolved* by performing multi-step aggregation within a single layer, since attention at one layer reflects only one-hop similarity, whereas multi-hop paths require information to actually propagate through intermediate tokens, which happens *across* layers.

**Intra-layer walks on full attention are prohibitively expensive, but sketching makes them tractable.** The natural solution is to perform a walk on the attention matrix *within* each layer to capture multi-hop dependencies: computing powers $A^s$ reveals which tokens are connected through multi-hop paths at that layer. However, this is prohibitively expensive, computing the full $n \times n$ dense attention matrix negates any efficiency gains from sparse attention. Our key insight is that we can perform a *blockwise* walk instead: by aggregating tokens into blocks via *token-space sketching* and compressing features via *feature-space sketching*, we reduce the intra-layer walk to $b \times b$ block matrices where $b \ll n$. This makes multi-hop aggregation tractable at inference time without retraining.

**We propose Sketch&Walk: sketch to estimate, walk to**

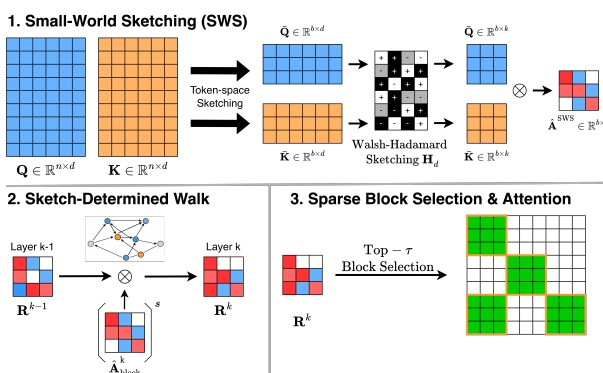

*Figure 1.* Overview of Sketch&Walk. (1) Queries and keys are sketched with Small-World Sketching to obtain lightweight block-level attention estimates. (2) These estimates are accumulated across layers with Sketch-Determined Walk to approximate cross-layer attention influence. (3) The resulting walk scores are used to select top-$\tau$ blocks for sparse attention.

**aggregate.** We introduce Sketch&Walk Attention, which captures multi-hop dependencies through a two-stage approach that operates entirely at inference time. First, **Small-World Sketching (SWS)** efficiently estimates block-level attention via token-space sketching and feature-space sketching, reducing multiplication complexity from $O(B^2 d)$ to $O(k)$ per block pair. Second, **Sketch-Determined Walk** uses these sketched attention estimates to perform a walk *at inference time per layer*: $R^k = R^{k-1}(\hat{\mathbf{A}}^k_{\text{block}})^s$. **This per-layer walk during inference is our key novelty**, enabling multi-hop aggregation *within* each layer's forward pass, rather than relying on depth-wise composition of attention layers. The *single-layer* walk state accumulates multi-hop paths connecting query and key blocks across the transformer depth. By selecting sparse attention targets from the walk state rather than one-hop scores, Sketch&Walk retains blocks that are important that can only be identified with composition of attention layers. We establish provable guarantees showing that Sketch&Walk recovers the correct sparse attention pattern with high probability.

**Sketch&Walk achieves near-lossless accuracy with substantial inference speedups.** Across multiple model scales and a broad range of tasks, Sketch&Walk maintains near-lossless accuracy at 80% sparsity and can slightly outperform dense attention in some settings. The same inference-time mechanism applies uniformly to both the prefill and decode phases. Combined with custom Triton kernels, Sketch&Walk delivers consistent speedups that grow with context length, achieving up to $4.7\times$ end-to-end attention speedup over FlashAttention-2 at 96K context.[1] Our contributions include:

- We identify a fundamental limitation of per-layer, one-hop sparse attention selection: it might fail to select tokens

---

[1]Our code is available at https://github.com/Escanord/SketchWalk

involved in multi-hop dependencies (induced by attention composition) and required in later layers.

- We propose Sketch&Walk, a training-free, low-overhead sparse attention method that applies to both prefill and decode phases of the inference process, capturing multi-hop token dependencies through Small-World Sketching and Sketch-Determined Walk.

- We establish theoretical approximation guarantees of Sketch&Walk and empirically demonstrate Sketch&Walk can achieve up to $4.7\times$ end-to-end attention speedup while preserving near-lossless accuracy across long-context benchmarks.

## 2. Sketch and Walk

We introduce **Sketch&Walk Attention**, based on **Small-World Sketching (SWS)** followed by a **Sketch-Determined Walk**. SWS applies *token-space sketching* via block-level aggregation and *feature-space sketching* via Hadamard projections to estimate block-level attention scores, which are then used to guide the Sketch-Determined Walk operation over attention blocks. The name reflects the key insight that small-world sketching induces a small-world network structure, where any token can reach another through a small number of high-importance block transitions, mirroring the classical small-world phenomenon in social networks. See Figure 1 for a complete overview of Sketch&Walk algorithm.

### 2.1. Preliminaries and Notation

Let $\mathbf{Q}, \mathbf{K} \in \mathbb{R}^{n \times d}$ denote the query and key matrices with $n$ tokens and head dimension $d$. We partition these into $b = \lceil n/B \rceil$ blocks of size $B$. For block $i$, let $\mathbf{Q}^{(i)}, \mathbf{K}^{(i)} \in \mathbb{R}^{B \times d}$ denote the corresponding sub-matrices.

**Definition 2.1** (Block Attention Score)**.** The true block attention score between query block $i$ and key block $j$ is defined as:

$$A_{ij}^{\text{true}} = \frac{1}{\sqrt{d}} \frac{1}{B^2} \sum_{s=1}^{B} \sum_{t=1}^{B} \mathbf{Q}^{(i)}[s,:] \cdot \mathbf{K}^{(j)}[t,:]^{\top}.$$

### 2.2. Small-World Sketching

**Small-World Sketching (SWS).** We formalize *Small-World Sketching* as a two-stage sketching operator that constructs a low-dimensional representation of token interactions for efficient block-level attention estimation.

- **Token-space sketching (block aggregation).** Given a sequence of token representations partitioned into blocks of size $B$, the $i$-th block representation is defined as

$$\bar{\mathbf{q}}_i \triangleq \frac{1}{B} \sum_{t=1}^{B} \mathbf{Q}^{(i)}[t,:], \quad \bar{\mathbf{k}}_i \triangleq \frac{1}{B} \sum_{t=1}^{B} \mathbf{K}^{(i)}[t,:] \in \mathbb{R}^d,$$

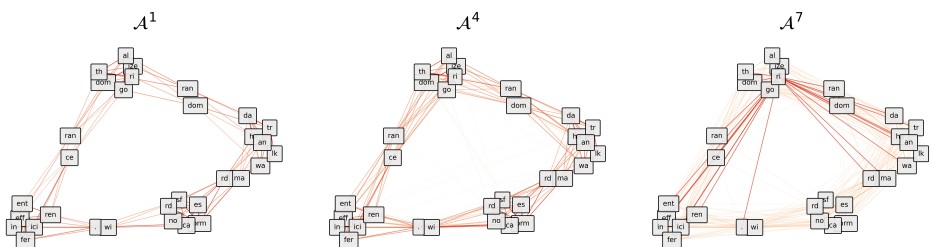

*Figure 2.* Visualization of attention matrices from a layer of the Llama-3.1-8B-Instruct. Each node corresponds to a token, and edge intensity reflects the magnitude of the attention score. $A^1$ (left) shows direct attention scores. Higher-order compositions of attention are shown by $A^4$ (middle) and $A^7$ (right). While $A^1$ captures only direct interactions, higher powers approximate the influence induced by repeated attention composition, reflecting attention that becomes strong in deeper layers.

which reduces the original sequence of $n$ tokens to $b = \lceil n/B \rceil$ block-level vectors.

- **Feature-space sketching (Hadamard projection).** Each block representation mapped to a lower-dimensional feature space via a randomized Hadamard projection,

$$\tilde{\mathbf{q}}_i \triangleq \bar{\mathbf{q}}_i \mathbf{H}_d \in \mathbb{R}^k, \qquad \tilde{\mathbf{k}}_i \triangleq \bar{\mathbf{k}}_i \mathbf{H}_d \in \mathbb{R}^k,$$

where $\mathbf{H}_d \in \mathbb{R}^{d \times k}$ is a Hadamard matrix and $k \ll d$ denotes the sketch dimension. The orthogonality of the Hadamard transform ensures that similarity estimates in the transformed space remain consistent with those in the original feature space.

Together, token-space and feature-space sketching produce a compact block-level sketch that supports efficient estimation of inter-block attention scores. In practice, we implement custom CUDA kernels for block aggregation and fast Hadamard transforms to reduce the runtime overhead of Small-World Sketching. Under Small-World Sketching, the block-level attention matrix is estimated as:

$$\hat{\mathbf{A}}^{\text{SWS}} \triangleq \frac{\tilde{\mathbf{Q}}\,\tilde{\mathbf{K}}^\top}{\sqrt{k}} = \frac{(\bar{\mathbf{Q}}\mathbf{H}_d)\,(\bar{\mathbf{K}}\mathbf{H}_d)^\top}{\sqrt{k}},$$

This reduces the complexity of computing block scores from $O(B^2 d)$ to $O(k)$ per block pair.

### 2.3. Sketch-Determined Walk

After estimating block-level attention scores via *Small-World Sketching (SWS)*, Sketch&Walk applies a *Sketch-Determined Walk* to aggregate these estimates across transformer layers. The resulting walk state $R^k$ is updated at layer $k$ as:

$$R^0 = (\hat{\mathbf{A}}^0_{\text{block}})^s, \qquad R^k = R^{k-1}(\hat{\mathbf{A}}^k_{\text{block}})^s, \quad k > 0,$$

where $\hat{\mathbf{A}}^k_{\text{block}}$ denotes the sketched block attention matrix at layer $k$, and $s$ is a sparsity exponent.

The sketch-determined walk state $R^k$ aggregates block-level attention estimates from the current layer with those propagated from preceding layers, yielding a block-level score

matrix that encodes *inter-layer relationships* among tokens and reflects *block-wise groupings* of blocks according to their layerwise attention relationships. The top-$\tau$ key blocks for sparse attention are then selected according to the corresponding row of $R^k$.

We implement custom Triton kernels for sparse attention in both the prefill and decode phases to efficiently execute attention over the selected blocks. Algorithms 1 and 2 provide a detailed walkthrough of **Sketch&Walk Attention** for the prefill and decode phases, respectively.

### 2.4. Error Bounds and Approximation Analysis

We analyze the approximation properties of Sketch&Walk Attention and establish provable guarantees for both block selection and sparse attention performance. We show that Sketch&Walk provides accurate identification of important attention blocks, and the resulting sparse attention closely approximates dense attention. We begin by formalizing assumptions on token representations and attention distributions:

**Assumption 2.2** (Block Coherence)**.** Within each block $\mathcal{B}_i$, token embeddings exhibit **semantic coherence**: tokens within a block share similar contextual representations. Formally, for query block $i$, each token $\mathbf{q}_t^{(i)}$ can be decomposed as:

$$\mathbf{q}_t^{(i)} = \boldsymbol{\mu}_i + \boldsymbol{\epsilon}_t, \quad \text{where } \mathbb{E}[\boldsymbol{\epsilon}_t] = \mathbf{0}, \text{ Var}(\boldsymbol{\epsilon}_t) \leq \sigma^2 \mathbf{I}$$

with intra-block variance $\sigma^2 \ll \|\boldsymbol{\mu}_i\|^2$. This assumption is empirically justified by the observation that consecutive tokens in natural language often belong to the same semantic unit (phrase, clause, or sentence), and positional encodings induce smooth variations within local windows.

**Assumption 2.3** (Heavy-Tailed Block Attention Distribution)**.** The distribution of block-level attention scores follows a heavy-tailed pattern: for each query block $i$, there exists a small subset $\mathcal{S}_i^* \subset \{1, \ldots, b\}$ with $|\mathcal{S}_i^*| = \tau \ll b$ such that: $\sum_{j \in \mathcal{S}_i^*} \text{softmax}(A_{ij}^{\text{true}}) \geq 1 - \eta$, for small $\eta > 0$ (typically $\eta < 0.1$). This reflects the finding that attention in LLMs concentrates on semantically relevant positions, with most blocks receiving negligible attention mass.

*Table 1.* Prefill phase performance comparison on LongBench. $\text{AVG}^{\overline{pc}}$ excludes PassageCount.

| | 2wikimqa | gov-report | hotpot-qa | lcc | multifieldqa-en | multinews | musique | narrativeqa | passage-count | passage-retrieval | qasper | qmsum | repobench-p | samsum | trec | triviaqa | AVG | $\text{AVG}^{\overline{pc}}$ |
|---|---|---|---|---|---|---|---|---|---|---|---|---|---|---|---|---|---|---|
| **Llama3.1-8B-Instruct** | | | | | | | | | | | | | | | | | | |
| Dense | 45.62 | 34.77 | 55.40 | 55.13 | 55.97 | 26.90 | 29.41 | 30.05 | 10.00 | 99.00 | 44.67 | 25.14 | 47.79 | 43.24 | 73.00 | 91.16 | 47.95 | 50.48 |
| FlexPrefill | 34.25 | 33.52 | 54.86 | 54.11 | 54.23 | 26.71 | 28.95 | 27.62 | 1.00 | 63.50 | 39.21 | **25.97** | **54.65** | 43.26 | 71.50 | 89.00 | 43.90 | 46.76 |
| MInference | 46.39 | 33.76 | 54.18 | **55.43** | 54.75 | 26.64 | 27.18 | 28.60 | 4.25 | 97.50 | 43.30 | 25.62 | 50.97 | 43.27 | **72.50** | 90.40 | 47.17 | 50.03 |
| SpargeAttn | 44.15 | 34.39 | 54.86 | 48.62 | 54.18 | 26.88 | 28.63 | 29.95 | 4.50 | 98.00 | 42.74 | 24.88 | 44.62 | 43.87 | 71.00 | 90.24 | 46.34 | 49.13 |
| XAttention | 45.14 | 34.86 | 55.07 | 53.76 | **55.49** | 26.83 | 30.81 | 30.22 | 3.50 | 96.00 | **44.57** | 25.63 | 47.66 | 43.44 | 72.00 | 91.81 | 47.30 | 50.22 |
| PBS-Attn | **47.89** | **34.89** | 56.51 | 52.64 | 53.36 | 26.81 | 30.94 | **32.18** | 1.86 | **99.00** | 42.24 | 25.47 | 46.85 | 43.12 | 72.00 | 91.07 | 47.30 | 50.33 |
| Sketch&Walk | 47.26 | 34.39 | **56.69** | 52.76 | 55.30 | **26.92** | 31.27 | 29.61 | 6.16 | 99.00 | 44.13 | 25.26 | 47.88 | **43.99** | 70.00 | **92.02** | 47.67 | 50.43 |
| **Llama3.2-1B-Instruct** | | | | | | | | | | | | | | | | | | |
| Dense | 29.34 | 29.49 | 30.26 | 30.29 | 41.29 | 25.96 | 14.61 | 18.59 | 3.14 | 5.00 | 21.05 | 21.46 | 25.44 | 39.26 | 62.00 | 78.89 | 29.75 | 31.53 |
| FlexPrefill | 21.76 | 29.24 | 29.97 | **29.62** | 36.97 | 25.33 | 13.34 | 16.38 | 4.64 | 4.50 | 16.18 | 21.24 | 27.44 | 38.24 | 58.00 | 74.55 | 27.96 | 29.52 |
| MInference | 29.40 | 29.56 | 30.46 | 29.59 | 40.46 | 25.06 | 14.44 | 16.70 | 1.14 | 5.00 | 20.44 | 21.64 | 26.81 | **39.42** | 61.50 | 76.21 | 29.24 | 31.11 |
| SpargeAttn | 30.11 | 29.61 | 24.75 | 24.95 | 35.47 | 26.17 | 11.13 | 16.34 | 3.05 | 4.50 | 17.19 | 20.80 | 32.04 | 37.87 | 54.50 | 74.76 | 27.70 | 29.35 |
| XAttention | 30.93 | **29.73** | 30.03 | 26.28 | 41.87 | 25.89 | **15.95** | 16.66 | 3.00 | 4.80 | 18.69 | **22.00** | 33.33 | 39.30 | 61.50 | 79.05 | 30.06 | 31.87 |
| PBS-Attn | **31.43** | 29.42 | 30.68 | 27.67 | **42.09** | **26.27** | 13.71 | 18.59 | **5.00** | **5.25** | 17.64 | 21.94 | 33.09 | 38.40 | 60.00 | **79.10** | 30.02 | 31.69 |
| Sketch&Walk | 29.96 | 29.68 | **31.08** | 29.58 | 41.49 | 25.73 | 15.54 | **18.67** | 2.64 | 5.00 | 20.44 | 21.82 | **33.54** | 39.04 | 60.00 | 77.59 | 30.11 | 31.94 |
| **Qwen2-7B-Instruct** | | | | | | | | | | | | | | | | | | |
| Dense | 43.71 | 32.12 | 43.99 | 48.64 | 47.31 | 24.39 | 25.26 | 23.82 | 5.50 | 69.00 | 46.74 | 22.95 | 47.66 | 33.59 | 55.00 | 83.97 | 40.85 | 43.21 |
| FlexPrefill | **43.77** | 32.22 | **43.45** | 47.17 | 46.73 | 24.61 | 24.48 | 25.20 | 5.50 | 61.00 | **47.47** | 23.19 | 47.12 | 34.22 | 61.00 | 85.68 | 40.80 | 43.15 |
| Sketch&Walk | 42.67 | **33.86** | 42.49 | **48.81** | **47.29** | **25.87** | **24.92** | **25.27** | 5.50 | **67.00** | 45.66 | **23.92** | **55.61** | **46.31** | **77.00** | **89.07** | **43.83** | **46.38** |
| **Qwen3-8B** | | | | | | | | | | | | | | | | | | |
| Dense | 41.96 | 33.72 | 58.50 | 56.06 | 53.77 | 25.09 | 34.73 | 26.83 | 3.50 | 100.00 | 48.33 | 23.94 | 55.45 | 44.21 | 71.50 | 90.71 | 48.02 | 50.99 |
| SpargeAttn | 40.95 | **33.79** | **56.43** | **54.34** | **51.93** | **24.68** | 30.81 | 24.80 | 2.00 | **99.50** | 45.13 | **23.65** | 52.48 | **44.57** | **71.00** | 90.71 | 46.67 | 49.65 |
| Sketch&Walk | **41.14** | 33.08 | 55.52 | 53.67 | 49.95 | 24.47 | **34.50** | **26.49** | 2.00 | 99.00 | **45.73** | 23.55 | **52.97** | 43.84 | 69.50 | **91.45** | **46.68** | **49.66** |

Under these assumptions, we show that token-space sketching via block aggregation acts as a subspace embedding, yielding *accurate block-level representations with high probability*. Meanwhile, feature-space sketching via the Subsampled Randomized Hadamard Transform *preserves block-level inner products up to small error* when the sketch dimension is sufficiently large.

**Lemma 2.4.** (Token-space Sketching: Subspace Embedding via Block Averaging). *Under Assumption 2.2, the block average $\bar{\mathbf{q}}_i = \frac{1}{B} \sum_{t=1}^{B} \mathbf{q}_t^{(i)}$ satisfies:*

$$\|\bar{\mathbf{q}}_i - \boldsymbol{\mu}_i\|_2 \leq \frac{\sigma \sqrt{d \log(1/\delta)}}{\sqrt{B}}$$

*with probability at least $1 - \delta$.*

**Theorem 2.5.** (Feature-space Sketching: Hadamard Transform Guarantee). *Let $\mathbf{H}_d \in \mathbb{R}^{d \times k}$ be the Subsampled Randomized Hadamard Transform (SRHT). For any fixed vectors $\mathbf{u}, \mathbf{v} \in \mathbb{R}^d$ and $\epsilon \in (0,1)$, if $k = \Omega(\epsilon^{-2} \log(b/\delta))$ where $b$ is the number of blocks, then with probability at least $1 - \delta$:*

$$\left| (\mathbf{u}^\top \mathbf{H}_d)(\mathbf{H}_d^\top \mathbf{v}) - \mathbf{u}^\top \mathbf{v} \right| \leq \epsilon \|\mathbf{u}\|_2 \|\mathbf{v}\|_2$$

Given sufficient sketch dimension and block size, the top-$\tau$ blocks selected using sketched attention scores coincide with those selected by full attention with high probability:

**Lemma 2.6.** (SWS Top-$\tau$ Selection under Softmax). *Under Assumptions 2.2 and 2.3, let $\mathcal{S}^*$ be the true top-$\tau$ blocks based on $\text{softmax}(A_{ij}^{true}/\sqrt{d})$, and let $\hat{\mathcal{S}}$ be the blocks selected by Sketch&Walk using $\text{softmax}(\hat{A}_{ij}^{SWS}/\sqrt{k})$. If the sketching dimension satisfies:*

$$k \geq \frac{C \log(b/\delta)}{\gamma^2} \cdot \max_{i,j} \|\bar{\mathbf{q}}_i\|^2 \|\bar{\mathbf{k}}_j\|^2$$

*and the block size satisfies $B \geq C'\sigma^2 d \log(1/\delta)/\gamma^2$, then $\mathcal{S}^* = \hat{\mathcal{S}}$ with probability at least $1 - 2\delta$.*

Finally, under the heavy-tailed attention assumption, restricting attention to the selected top-$\tau$ blocks yields an output that closely approximates dense attention, with error controlled by the tail mass and the quality of token-space aggregation:

**Lemma 2.7.** (Attention Output Approximation). *Under Assumptions 2.2–2.3, let $\mathbf{O}^{full}$ be the output of full attention and $\mathbf{O}^{Sketch\&Walk}$ be the output using Sketch&Walk sparse attention with top-$\tau$ blocks. Then:*

$$\|\mathbf{O}^{Sketch\&Walk} - \mathbf{O}^{full}\|_F \leq \eta \|\mathbf{V}\|_F + O\left(\frac{\tau\sigma}{\sqrt{B}}\right) \|\mathbf{V}\|_2 \tag{1}$$

*where $\eta$ is the tail mass from Assumption 2.3.*

Detailed proofs of the above lemmas and theorems are provided in Appendix B.3.1.

*Table 2.* End-to-End performance of Sketch&Walk on RULER.

| Model | Method | Context Length | | | | | |
|---|---|---|---|---|---|---|---|
| | | 4K | 8K | 16K | 32K | 64K | Average |
| **L3.1 8B** | Dense | 95.31 | 95.31 | 94.06 | 89.69 | 82.81 | 91.44 |
| | Sketch&Walk | 93.64 | 95.00 | 93.13 | 87.50 | 81.50 | 90.15 |

## 3. Why Sketch&Walk for Sparse Attention?

Many existing sparse attention methods such as (Tang et al., 2024; Yan et al., 2025; Lai et al., 2025) select key blocks using layer-wise relevance scores that approximate direct attention. This design introduces two implicit assumptions: (i) that *block relevance can be determined from direct attention scores*, and (ii) that *block selection can be performed independently at each layer*. While this might be effective in some tasks, we believe these assumptions break down in deep transformer architecture, i.e. LLMs.

From a theoretical perspective, the failure of direct-score selection follows from the properties of inner-product–based relevance measures. Attention scores are defined by inner products and therefore induce a non-metric similarity measure. As shown in Remark B.9, the inner product violates the triangle inequality. Thus, the existence of a block sequence $i \to j \to k$ with large pairwise relevance scores does not imply a large direct relevance score between $i$ and $k$. Under per-layer block selection, a query block $i$ will not select a key block $k$ when their relevance is expressed through intermediate blocks. Such relevance becomes apparent only after attention is accumulated in deeper layers (Cai et al., 2024; Abnar & Zuidema, 2020). However, block $k$ was excluded from sparse attention in earlier layers, so this dependency cannot be recovered when it later becomes important.

This limitation is also evident empirically. We visualize attention matrices from a layer of the Llama-3.1-8B-Instruct model (Grattafiori et al., 2024) in Figure 2. The matrix $A^1$ corresponds to direct attention scores, which can be viewed as an oracle relevance metric commonly used by existing sparse attention methods for top-$k$ block selection. While $A^1$ captures only immediate interactions, higher-order compositions such as $A^4$ and $A^7$ gradually reveal structure that emerges through repeated composition of attention operators. Blocks that appear weakly connected under $A^1$ can receive substantial mass under higher powers, indicating their relevance in later layers.

The visualization raises a concrete question: how often does a block become important only after several additional layers of attention composition, and to what extent can a single-layer score anticipate this? To answer it, we examine 50 samples from the multi-hop subsets of LongBench (Bai et al., 2024) (MuSiQue, 2WikiMQA, HotpotQA) on Llama-3.1-8B-Instruct, and at each layer $L$ form the set of KV blocks that enter the oracle top-$k$ at layer $L$ but were absent

$x$ layers earlier,

$$G(L, x) = \text{OracleTop-k}(L) \setminus \text{OracleTop-k}(L - x).$$

Any score computed at layer $L - x$ that aims to drive sparse attention at layer $L$ must already mark the blocks in $G(L, x)$ as important. We compare three such scores under a matched block budget: *Quest* (Tang et al., 2024), the per-token max $Q{\cdot}K$ score used by single-layer methods; *Sketch*, the sketched block-attention matrix $\hat{A}_{\text{block}}^{L-x}$ used inside Sketch&Walk but without cross-layer composition; and Sketch&Walk itself, i.e. the full walk state $R^{L-x}$. Recall against $G(L, x)$, averaged over layers 2–31, is reported in Table 3.

*Table 3.* Recall against the multi-hop target set $G(L, x)$, averaged over layers 2–31 and 50 samples from the multi-hop subsets of LongBench on Llama-3.1-8B-Instruct.

| Offset $x$ | Sketch&Walk | Sketch | Quest |
|---|---|---|---|
| 1 | **0.46** | 0.27 | 0.23 |
| 5 | **0.44** | 0.24 | 0.19 |
| 10 | **0.43** | 0.25 | 0.18 |

The ordering Sketch&Walk > Sketch > Quest is stable across every layer and every lookback we examined, with Sketch&Walk recovering nearly twice as many multi-hop blocks as Quest. The Quest-to-Sketch gap isolates the contribution of block-level aggregation, while the Sketch-to-Sketch&Walk gap isolates the contribution of cross-layer composition; both are substantial. Together with Figure 2, this confirms that a non-trivial fraction of the blocks needed at layer $L$ are not predictable from layer $L - x$ alone, and that accumulating attention across layers is what makes them recoverable.

These observations highlight a fundamental gap: *block relevance is not solely determined by the current layer's attention scores*. Instead, relevance emerges through the composition of attention across layers, and block selection for sparse attention must account for this accumulation in order to preserve information required by subsequent layers. This motivates selecting blocks based on higher-order attention structure rather than per-layer scores.

Sketch&Walk addresses this issue by explicitly accumulating attention across layers. At each layer, we estimate a block-level attention matrix $\hat{A}_{\text{block}}^k$ using Small-World Sketching and maintain a Sketch-Determined Walk state

$$R_0 = (\hat{A}_{\text{block}}^0)^s, \qquad R_k = R_{k-1}(\hat{A}_{\text{block}}^k)^s, \qquad (2)$$

which captures higher-order compositions of attention. As formalized in Lemma B.12, selecting top-$\tau$ blocks based on $R^k[i, j]$ identifies blocks whose importance emerges consistently across layers, rather than those that are locally salient

*Table 4.* Decoding phase performance comparison on LongBench. AVG$^{\overline{pc}}$ excludes PassageCount.

| | 2wikimqa | govreport | hotpotqa | lcc | multifieldqa-en | multinews | musique | narrativeqa | passage-count | passage-retrieval | qasper | qmsum | repobench-p | samsum | trec | triviaqa | AVG | AVG$^{\overline{pc}}$ |
|---|---|---|---|---|---|---|---|---|---|---|---|---|---|---|---|---|---|---|
| | | | | | | | **Llama3.1-8B-Instruct** | | | | | | | | | | | |
| Dense | 45.62 | 34.77 | 55.40 | 55.13 | 55.97 | 26.90 | 29.41 | 30.05 | 10.00 | 99.00 | 44.67 | 25.14 | 47.79 | 43.24 | 73.00 | 91.16 | 47.95 | 50.48 |
| Quest | 46.95 | **34.80** | 55.93 | 49.96 | **55.98** | **27.32** | 30.89 | 28.79 | 8.03 | 99.00 | 42.21 | 24.19 | 45.12 | 42.64 | 69.00 | 89.90 | 46.92 | 49.51 |
| Adamas | 45.81 | 34.72 | 56.07 | 54.20 | 56.20 | 26.89 | 31.20 | 30.09 | 8.00 | 99.00 | 43.84 | 25.05 | 46.96 | 43.14 | **73.00** | 91.29 | 47.84 | 50.49 |
| Sketch&Walk | **47.11** | 32.39 | **57.49** | 54.94 | 55.73 | 25.55 | **33.16** | **30.69** | 8.75 | 99.00 | **44.44** | **25.47** | **47.78** | 43.39 | 70.50 | **91.40** | 48.00 | 50.60 |
| | | | | | | | **Llama3.2-1B-Instruct** | | | | | | | | | | | |
| Dense | 29.34 | 29.49 | 30.26 | 30.29 | 41.29 | 25.96 | 14.61 | 18.59 | 3.14 | 5.00 | 21.05 | 21.46 | 25.44 | 39.26 | 62.00 | 78.89 | 29.75 | 31.53 |
| Quest | 28.93 | 26.61 | 30.14 | 31.50 | 38.95 | 24.54 | 14.51 | 17.18 | 2.50 | 5.00 | 19.79 | 20.39 | 33.48 | 38.37 | 60.00 | 71.58 | 28.97 | 30.73 |
| Adamas | 29.45 | **29.41** | 29.06 | **30.70** | 40.08 | 23.86 | 15.10 | 17.94 | **3.14** | 5.00 | 19.34 | 20.95 | 34.03 | 37.63 | **62.00** | 77.89 | 29.72 | 31.50 |
| Sketch&Walk | **30.14** | 28.11 | **30.79** | 29.35 | **40.38** | 24.91 | **15.54** | **19.30** | 3.14 | **5.75** | 20.18 | **21.78** | 36.02 | **38.79** | 61.50 | **79.81** | **30.34** | **32.16** |

in a single layer. These blocks minimize reconstruction error with respect to the full transformer attention pattern.

By selecting blocks from the accumulated relevance scores in $R^k$, Sketch&Walk mitigates the limitations of layer-wise selection and enables sparse attention to better approximate dense multi-layer attention behavior.

*Table 5.* Prefill phase performance comparison on RULER.

| Model | Method | Context Length | | | | | |
|---|---|---|---|---|---|---|---|
| | | 4K | 8K | 16K | 32K | 64K | Average |
| **L3.1 8B** | Dense | 95.31 | 95.31 | 94.06 | 89.69 | 82.81 | 91.44 |
| | FlexPrefill | 95.31 | 92.50 | 93.75 | 88.13 | 80.31 | 90.00 |
| | MInference | 92.65 | **95.00** | 93.12 | 86.87 | **82.50** | 90.03 |
| | Sketch&Walk | **96.56** | 94.06 | **94.06** | **89.63** | 82.50 | **91.35** |
| **L3.2 1B** | Dense | 74.38 | 68.44 | 65.31 | 66.88 | 66.56 | 68.31 |
| | FlexPrefill | **74.06** | 63.75 | 60.63 | 61.88 | 55.94 | 63.25 |
| | MInference | 73.44 | **69.06** | 64.69 | 64.38 | 62.71 | 66.86 |
| | Sketch&Walk | **74.06** | 68.44 | **66.06** | 65.62 | 63.75 | 67.59 |

## 4. Experiments

### 4.1. Settings

**Models.** To maximize coverage of models supported by all baselines, we evaluate Sketch&Walk on widely used long-context instruction-tuned models of different scales and families: Llama-3.1-8B-Instruct (Grattafiori et al., 2024), Llama-3.2-1B-Instruct (Meta, 2024), Qwen2-7B-Instruct (Yang et al., 2024), and Qwen3-8B (Yang et al., 2025). We use the default chat template for all instruct models in subsequent experiments. Following Tang et al. (2024), we do not apply Sketch&Walk to the first two layers, as these layers typically exhibit low achievable sparsity.

**Datasets.** We evaluate methods on two benchmarks that pose complementary challenges for long-context understanding: (i) LongBench (Bai et al., 2024), a diverse benchmark covering question answering, reasoning, summarization, and code understanding tasks with input lengths up to tens of thousands of tokens; and (ii) RULER (Hsieh et al., 2024), a synthetic diagnostic benchmark designed to stress-test a model's ability to retrieve and utilize sparse, position-sensitive information within very long contexts.

**Implementation Details.** All experiments are conducted on single NVIDIA H100 and H200 GPUs. We implement a custom inference pipeline in PyTorch to support efficient attention computation over long-context inputs. Our implementation leverages Triton (Tillet et al., 2019) to optimize GPU kernel performance. Following common practice in block-sparse attention, we always retain the first and last key blocks for each query block. All evaluations are performed using greedy decoding to ensure result consistency. Unless otherwise specified, Sketch&Walk uses a sketching dimension of 64 and sparsity exponent of 8.

**Baselines.** We compare Sketch&Walk against dense attention and competitive sparse attention methods. For prefill optimization, we evaluate against MInference (Jiang et al., 2024), FlexPrefill (Lai et al., 2025), SpargeAttn (Zhang et al., 2025), XAttention (Xu et al., 2025), and PBS-Attn (Wang et al., 2026). For decode optimization, we compare against QUEST (Tang et al., 2024) and Adamas (Yan et al., 2025). For each baseline, we adopt the best configuration reported in its respective paper at the same sparsity level to ensure a fair comparison.

### 4.2. Accuracy Evaluation

**Prefill Phase Comparison.** Tables 1 and 5 report the prefill-phase performance of Sketch&Walk compared against prefill-optimized sparse attention methods on LongBench and RULER, respectively. Evaluations are conducted at an 80% attention sparsity level. Across both benchmarks, spanning short to long contexts (4K to 64K tokens) on RULER and diverse real-world tasks on LongBench, Sketch&Walk achieves the best overall performance among all compared methods. Notably, Sketch&Walk consistently preserves model accuracy across a wide range of context lengths and, in some settings, even improves performance (e.g. about 3% on overall performance of Qwen2 on LongBench) while accelerating prefill computation.

**Decode Phase Comparison.** Tables 4 and 6 report the decode-phase performance of Sketch&Walk compared against decode-optimized sparse attention methods on Long-Bench and RULER, respectively, under an 80% attention sparsity setting. Across both benchmarks, Sketch&Walk achieves competitive performance relative to existing baselines, closely matching dense attention and, in some cases, surpassing it.

*Table 6.* Decoding phase performance comparison on RULER.

| Model | Method | Context Length | | | | | |
|---|---|---|---|---|---|---|---|
| | | 4K | 8K | 16K | 32K | 64K | Average |
| **L3.1 8B** | Dense | 95.31 | 95.31 | 94.06 | 89.69 | 82.81 | 91.44 |
| | Quest | 91.88 | 92.81 | 92.81 | 87.50 | 80.31 | 89.06 |
| | Adamas | 95.31 | 94.06 | **94.06** | **89.06** | 81.19 | 90.74 |
| | Sketch&Walk | **95.94** | **96.19** | 93.13 | 88.63 | **81.50** | 91.08 |

**End-to-End Performance.** Tables 7 and 2 show the end-to-end performance of Sketch&Walk on LongBench and RULER, respectively, when applied end-to-end across both the prefill and decode stages. On LongBench, Sketch&Walk closely matches dense attention across a wide range of tasks, with comparable average accuracy and minimal degradation under an 80% sparsity setting. On RULER, Sketch&Walk maintains strong performance across context lengths from 4K to 64K tokens, exhibiting only a modest average accuracy drop relative to dense attention. Overall, these results demonstrate that Sketch&Walk preserves end-to-end model quality while benefiting from sparsity-induced efficiency gains throughout both phases of the full inference process.

### 4.3. Efficiency Evaluation

We measure the wall-clock speedup of a single attention layer relative to the dense FlashAttention-2 (Dao et al., 2022) baseline. All measurements are taken with Llama-3.1-8B-Instruct (32 query heads, 8 KV heads, head dimension 128) on a single NVIDIA H200 GPU in bf16. Timings are averaged over at least 20 iterations after 5 warmup runs, and Sketch&Walk is configured with keeping the top 10% of key blocks per query (i.e., 90% attention sparsity). Sparse-attention baselines are grouped by the phase they natively support: prefill-side (MInference (Jiang et al., 2024), Flex-Prefill (Lai et al., 2025)), and decode-side (Quest (Tang et al., 2024), Adamas (Yan et al., 2025)). For each setting we compare against the methods that apply, using the best per-method configuration reported in the literature at matching sparsity.

**Prefill phase.** Figure 3 reports per-layer prefill speedup as a function of sequence length. Sketch&Walk crosses the dense baseline after 8K tokens and grows monotonically with context length, reaching 3.84× at 96K. FlexPrefill is competitive while MInference, in contrast, has a fixed per-call orchestration overhead that does not amortize until the

context is long enough for dense FA2 itself to dominate.

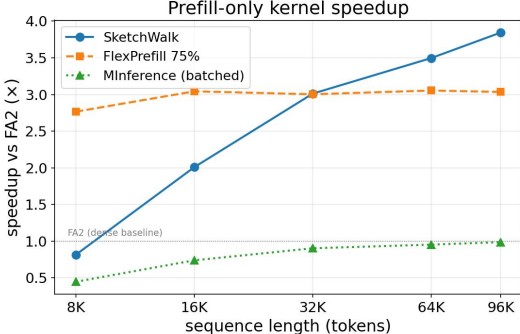

*Figure 3.* Single-layer prefill speedup over FlashAttention-2 on Llama-3.1-8B-Instruct at 90% sparsity.

**Decode phase.** Figure 4 reports per-step decode speedup as a function of the KV-cache length. Sketch&Walk's per-step latency stays roughly flat as the cache grows, while dense attention scales linearly, yielding a speedup that grows monotonically with context length and reaches 4.91× at 96K. Quest provides a steady ∼ 2.5× from 32K onward: Sketch&Walk trails Quest at ≤ 32K and surpasses it from 64K onward. Adamas, under the configuration documented in its paper, did not exceed dense FA2 in our setup.

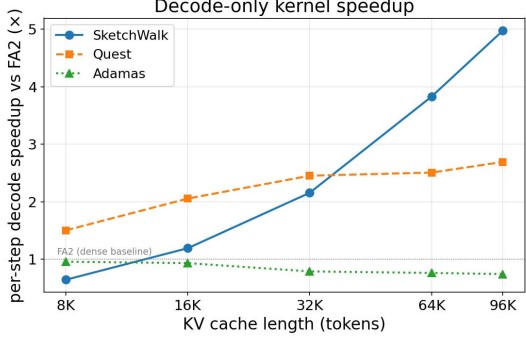

*Figure 4.* Single-layer decode speedup over FlashAttention-2 at varying KV-cache lengths.

**End-to-end (prefill + decode).** Figure 5 reports the combined speedup of one full prefill followed by 50 decode steps. Sketch&Walk is the only method that natively sparsifies both phases; for the other baselines we report the most favorable composition, namely the method's sparse phase combined with dense FA2 for the phase it does not support. Sketch&Walk delivers the largest end-to-end speedup at long context, reaching 4.69× at 96K, because its savings compound across both phases. Decode-only methods (Quest, Adamas) lose ground as context grows because the dense prefill increasingly dominates total time; prefill-only methods (FlexPrefill, MInference) gain only modestly because the savings are not compounded into decoding.

**Kernel analysis.** To verify that the speedups above are not offset by score estimation or the walk update, we decom-

*Table 7.* End-to-End performance of Sketch&Walk on LongBench. Sketch&Walk is evaluated at 80% sparsity level. $AVG^{\overline{pc}}$ excludes PassageCount.

| | 2wiki mqa | gov report | hotpot qa | lcc | multifieldqa en | multi news | musique | narrative qa | passage count | passage retrieval | qasper | qmsum | repobench | samsum | trec | trivia qa | AVG | $AVG^{\overline{pc}}$ |
|---|---|---|---|---|---|---|---|---|---|---|---|---|---|---|---|---|---|---|
| **Llama3.1-8B-Instruct** | | | | | | | | | | | | | | | | | | |
| Dense | 45.62 | 34.77 | 55.40 | 55.13 | 55.97 | 26.90 | 29.41 | 30.05 | 10.00 | 99.00 | 44.67 | 25.14 | 47.79 | 43.24 | 73.00 | 91.16 | 47.95 | 50.48 |
| Sketch&Walk | 45.54 | 33.22 | 56.51 | 53.78 | 54.53 | 25.96 | 33.11 | 31.18 | 6.53 | 98.00 | 44.14 | 25.17 | 48.63 | 43.69 | 70.00 | 92.08 | 47.63 | 50.37 |
| **Llama3.2-1B-Instruct** | | | | | | | | | | | | | | | | | | |
| Dense | 29.34 | 29.49 | 30.26 | 30.29 | 41.29 | 25.96 | 14.61 | 18.59 | 3.14 | 5.00 | 21.05 | 21.46 | 25.44 | 39.26 | 62.00 | 78.89 | 29.75 | 31.53 |
| Sketch&Walk | 29.54 | 28.34 | 31.01 | 29.35 | 40.97 | 24.61 | 14.11 | 17.98 | 3.00 | 5.30 | 19.67 | 21.56 | 36.02 | 38.79 | 61.50 | 79.81 | 30.10 | 31.90 |
| **Qwen2-7B-Instruct** | | | | | | | | | | | | | | | | | | |
| Dense | 43.71 | 32.12 | 43.99 | 48.64 | 47.31 | 24.39 | 25.26 | 23.82 | 5.50 | 69.00 | 46.74 | 22.95 | 47.66 | 33.59 | 55.00 | 83.97 | 40.85 | 43.21 |
| Sketch&Walk | 42.14 | 31.59 | 42.55 | 48.27 | 46.92 | 24.96 | 23.73 | 24.69 | 5.50 | 65.00 | 45.59 | 23.28 | 55.26 | 46.56 | 76.00 | 89.51 | 43.22 | 45.74 |

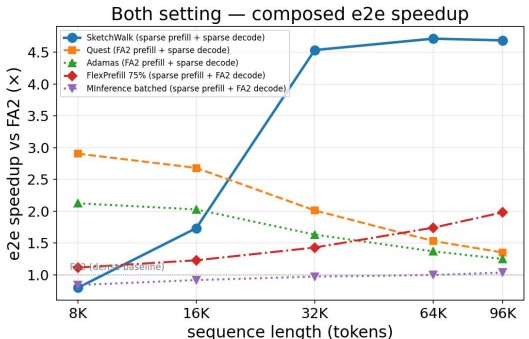

*Figure 5.* End-to-end speedup of 1 prefill + 50 decode steps over a fully dense FA2 pipeline.

pose the per-layer latency of Sketch&Walk into Hadamard sketching, sketch-determined walk, and sparse attention, and compare against dense FA2 at sequence lengths 16K, 32K, and 64K under token budgets of 512, 1024, and 2048 kept tokens per query (Figure 6). Two observations hold across all three context lengths: (i) the Hadamard sketching and walk-update components together account for only a small constant fraction of the total Sketch&Walk latency, with attention itself dominating the bar; and (ii) the gap between sparse and dense attention widens sharply as the sequence length grows, going from roughly 2× at 16K to nearly an order of magnitude at 64K. This is consistent with the end-to-end measurements in Figures 3–5 and confirms that the sketching and walk components do not erode the asymptotic benefits of block-sparse attention.

### 4.4. Ablation Studies

**Robustness to sketch dimension.** Figure 7(a) shows the effect of the Hadamard sketch dimension. Among 3 datasets, Sketch&Walk exhibits consistently strong performance over a wide range of sketch sizes. Even with small sketch dimensions (e.g., 16–32), accuracy remains close to that of larger sketch size. Sketch&Walk remains effective under aggressive dimensionality reduction.

**Sensitivity to walk degree.** Figure 7(b) studies the impact of the walk degree. Performance improves with increasing walk degrees and stabilizes thereafter, suggesting that only limited attention composition is needed in practice. Notably, increasing the walk degree does not introduce instability or degrade performance, indicating that the Sketch-Determined Walk behaves reliably across a broad range of settings.

**Robustness under varying sparsity.** Figure 7(c) reports results under different sparsity ratios. Sketch&Walk maintains near-lossless accuracy even at high sparsity levels. Sketch&Walk remains effective in highly constrained inference regimes and can achieve favorable accuracy–efficiency trade-offs.

## 5. Related Works

A large body of work has addressed the computational and memory bottlenecks of attention in long-context settings. These methods typically exploit natural sparsity in the attention mechanism, identified using inexpensive heuristics or approximations, to either reduce computation on less important token pairs or reduce the memory footprint of key–value (KV) caches. Prior approaches often target a specific stage of inference (prefill or decode), rely on auxiliary training, or introduce architectural constraints. In contrast, Sketch&Walk is a training-free sparse attention method that applies end-to-end to both prefill and decode phases. For more related works, please refer to Appendix A.

**Sparse attention for prefilling.** Several methods accelerate the prefill phase by restricting attention to a subset of interactions. Static sparsity patterns, such as Sparse Transformer (Child et al., 2019), Longformer (Beltagy et al., 2020), and BigBird (Zaheer et al., 2020), reduce complexity using fixed local or block-based layouts. Dynamic sparsity methods aim to adapt attention patterns to the input. MInference (Jiang et al., 2024) selects important interactions using a pre-determined sparsity pattern, while Flex-Prefill (Lai et al., 2025) leverages compiler-supported flexible block patterns. Although effective, these approaches often incur nontrivial pre-computation or scheduling over-

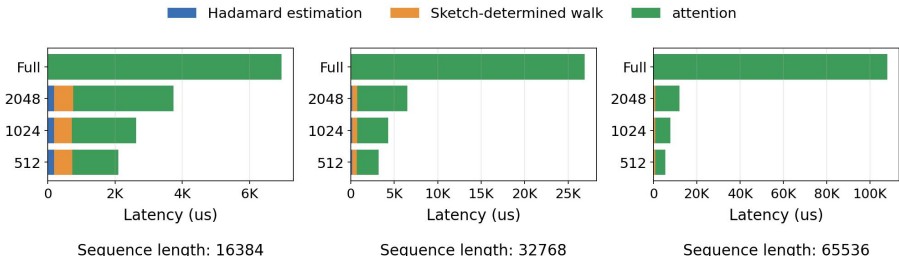

*Figure 6.* Sketch&Walk kernel breakdown at sequence lengths 16K, 32K, and 64K. Each panel compares dense FA2 ("Full") against Sketch&Walk at token budgets of 512, 1024, and 2048 kept tokens per query, with the latency split into Hadamard estimation, sketch-determined walk, and attention.

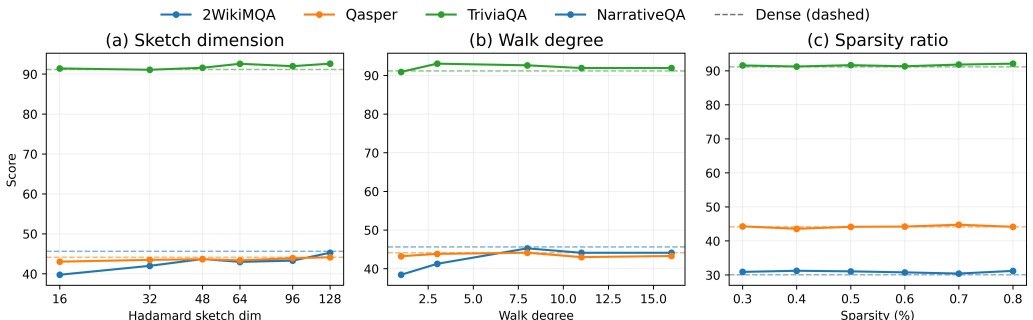

*Figure 7.* Ablation Studies on different parameters and sparsity level of Sketch&Walk

head. Training-based methods such as SeerAttention (Gao et al., 2024) introduce sparsity through learned gating mechanisms, improving efficiency at the cost of additional training and potentially reduced generalization.

**Efficient attention for decoding.** Another line of work focuses on reducing memory and computation during decoding by dynamically selecting, pruning, or compressing the KV cache. Methods such as $H_2O$ (Zhang et al., 2023), TOVA (Oren et al., 2024), and InfLLM (Xiao et al., 2024a) discard tokens based on query-dependent importance criteria. StreamingLLM (Xiao et al., 2024b) retains only initial and recent tokens to achieve stable latency and memory usage. More recent approaches, including Quest (Tang et al., 2024) and Rectified Sparse Attention (Sun et al., 2025), adaptively select tokens to maintain accuracy under high sparsity. SOCKET (Joshi et al., 2026) and MagicPig (Chen et al., 2025) employ principled, data-agnostic clustering to select salient keys for efficient sparse attention computation. These methods primarily target the decode phase.

Among these methods, Adamas (Yan et al., 2025) follows the close direction to Sketch&Walk. Both approaches leverage the Hadamard transform to obtain lightweight representations of queries and keys. However, Adamas uses the transformed representations directly as a distance-based criterion for token selection and operates in an independent, per-layer manner during decoding. In contrast, Sketch&Walk uses sketched representations to estimate attention score and explicitly accounts for cross-layer composition through the Sketch-Determined Walk. This design allows Sketch&Walk

to capture dependencies that emerge across layers and to apply uniformly to both prefill and decode phases.

## 6. Conclusion

We introduce Sketch&Walk, a training-free sparse attention method that applies uniformly to both the prefill and decode phases of LLM inference. By guiding sparse selection using sketched, walk-accumulated attention estimates across layers, Sketch&Walk achieves substantial inference speedups while preserving model quality. Empirically, Sketch&Walk maintains accuracy with minimal degradation, and in some cases yields improvements over dense attention, at impressive 80% sparsity. These results demonstrate that accounting for cross-layer attention composition enables effective and robust sparse attention for long-context inference. We discuss the practical limitations of Sketch&Walk in Appendix E.

## Acknowledgements

The work was supported by Ken Kennedy Generative AI cluster grants.

## Impact Statement

This paper presents work whose goal is to advance the field of Machine Learning. There are many potential societal consequences of our work, none, which we feel must be specifically highlighted here.

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

# A. More Related Works

**Alternative attention architectures.** Beyond sparse attention methods, several works propose alternative attention mechanisms or architectural changes. These include sliding-window attention (Beltagy et al., 2020), linear or gated attention (Qiu et al., 2025), and state-space models (Gu & Dao, 2024). More recent designs such as Native Sparse Attention (Yuan et al., 2025) and DeepSeek Sparse Attention (DeepSeek-AI, 2025) integrate sparsity at the architectural level. While effective in some regimes, these approaches typically require model modification or retraining. By contrast, Sketch&Walk is a post-training method that accelerates inference without architectural changes, proxy objectives, or additional training.

**Randomized Algorithms Drive Sparsity.** Randomized algorithms, particularly hashing algorithms and its variants (Broder, 1997; Broder et al., 1998; Charikar, 2002; Andoni et al., 2014; Andoni & Razenshteyn, 2015; Andoni et al., 2015; 2017; Li & Li, 2019a;b; 2021b;a; 2023; Zandieh et al., 2020; Zhang et al., 2018), have been instrumental in enabling sparsity-driven efficiency in machine learning systems. The MONGOOSE framework (Chen et al., 2021) introduced learnable hashing for efficient neural network training by dynamically sparsifying gradient computations. Building on maximum inner product (MaxIP) data structures, Xu et al. (2021) broke the linear iteration cost barrier for sparse conditional gradient methods. Guo et al. (2025) leveraged randomized empirical Fisher estimation to identify very sparse parameter subsets of LLMs, enabling memory-efficient zeroth-order fine-tuning with transferable static sparsity. More recently, Zen (Wang et al., 2025) leverages hierarchical hashing to achieve sparsity-driven data synchronization for distributed training. However, randomized sparsity often introduces an accuracy-efficiency tradeoff (Xu, 2025). To address this, soft prompts have been shown to recover the performance loss of randomized algorithm-driven sparse LLMs in a transferable manner (Xu et al., 2024), and SIRIUS (Zhou et al., 2024) introduced contextual sparsity with correction mechanisms for efficient inference.

# B. Theoretical Analysis

## B.1. Notation Summary

For ease of reference, we summarize the key mathematical notation used throughout this section in Table 8.

## B.2. Key Definitions and Assumptions

**Definition B.1** (Block Attention Score). The true block attention score between query block $i$ and key block $j$ is defined as:

$$A_{ij}^{\text{true}} = \frac{1}{B^2} \sum_{s=1}^{B} \sum_{t=1}^{B} \mathbf{Q}^{(i)}[s,:] \cdot \mathbf{K}^{(j)}[t,:]^{\top}$$

**Assumption B.2** (Block Coherence). Within each block $\mathcal{B}_i$, token embeddings exhibit **semantic coherence**: tokens within a block share similar contextual representations. Formally, for query block $i$, each token $\mathbf{q}_t^{(i)}$ can be decomposed as:

$$\mathbf{q}_t^{(i)} = \boldsymbol{\mu}_i + \boldsymbol{\epsilon}_t, \quad \text{where } \mathbb{E}[\boldsymbol{\epsilon}_t] = \mathbf{0}, \ \text{Var}(\boldsymbol{\epsilon}_t) \leq \sigma^2 \mathbf{I}$$

with intra-block variance $\sigma^2 \ll \|\boldsymbol{\mu}_i\|^2$. This assumption is empirically justified by the observation that consecutive tokens in natural language often belong to the same semantic unit (phrase, clause, or sentence), and positional encodings induce smooth variations within local windows.

**Assumption B.3** (Heavy-Tailed Block Attention Distribution). The distribution of block-level attention scores follows a heavy-tailed pattern: for each query block $i$, there exists a small subset $\mathcal{S}_i^* \subset \{1,\ldots,b\}$ with $|\mathcal{S}_i^*| = \tau \ll b$ such that:

$$\sum_{j \in \mathcal{S}_i^*} \text{softmax}(A_{ij}^{\text{true}}) \geq 1 - \eta$$

for small $\eta > 0$ (typically $\eta < 0.1$). This reflects the empirical finding that attention in LLMs concentrates on semantically relevant positions, with most blocks receiving negligible attention mass.

*Table 8.* Summary of notation used in theoretical analysis.

| Symbol | Description | Symbol | Description |
|---|---|---|---|
| *Dimensions, Indices & Basic Matrices* | | | |
| $n$ | Total number of tokens | $d$ | Head dimension |
| $B$ | Block size (tokens per block) | $b$ | Number of blocks, $b = \lceil n/B \rceil$ |
| $L$ | Number of transformer layers | $h$ | Number of attention heads |
| $k, r$ | Sketch dimension | $\tau$ | Top blocks for sparse attention |
| $\mathbf{Q}, \mathbf{K}, \mathbf{V}$ | Query, key, value matrices | $\mathbf{Q}^{(i)}, \mathbf{K}^{(i)}$ | Submatrices for block $i$ |
| $\bar{\mathbf{q}}_i, \bar{\mathbf{k}}_i$ | Block-averaged vectors | $\tilde{\mathbf{q}}_i, \tilde{\mathbf{k}}_i$ | Sketched block vectors |
| $\mathbf{q}_t^{(i)}$ | $t$-th token in query block $i$ | | |
| *Block Coherence Model & Sketching Transforms* | | | |
| $\boldsymbol{\mu}_i$ | Block centroid for block $i$ | $\boldsymbol{\epsilon}_t$ | Noise for token $t$ |
| $\sigma^2$ | Intra-block variance bound | $\mathbf{H}_d$ | SRHT matrix |
| $\mathbf{D}$ | Diagonal Rademacher matrix | $\mathbf{W}$ | Walsh-Hadamard matrix |
| $\mathbf{S}$ | Coordinate sampling matrix | | |
| *Block Attention & Sketch-Determined Walk* | | | |
| $A_{ij}^{\text{true}}$ | True block attention score | $\hat{A}_{ij}^{\text{SWS}}$ | Estimated block score (SWS) |
| $\hat{\mathbf{A}}_{\text{block}}^k$ | Sketched block attention at layer $k$ | $R^k$ | Walk state matrix at layer $k$ |
| $\mathbf{W}^k$ | $(\hat{\mathbf{A}}_{\text{block}}^k)^s$, exponentiated attention | $s$ | Sparsity exponent ($s > 1$) |
| *Block Selection Sets* | | | |
| $\mathcal{B}_i$ | The $i$-th block of tokens | $\mathcal{S}_i^*$ | True top-$\tau$ blocks for block $i$ |
| $\hat{\mathcal{S}}$ | Estimated top-$\tau$ blocks | $\mathcal{R}_i^L$ | Reachable blocks after $L$ layers |
| $\mathcal{C}_i$ | Consistently important blocks | | |
| *Attention Matrices & Outputs* | | | |
| $\mathbf{A}^{\text{full}}$ | Full (dense) attention matrix | $\mathbf{A}^{\text{sparse}}$ | Sparse attention matrix |
| $\mathbf{A}^{\text{oracle}}$ | Oracle sparse attention | $\mathbf{O}^{\text{full}}$ | Full attention output |
| $\mathbf{O}^{\text{Sketch\&Walk}}$ | Sketch&Walk output | $\mathbf{O}_\ell^{\text{sparse}}$ | Sparse output at layer $\ell$ |
| *Error, Probability & Markov Chain Parameters* | | | |
| $\eta$ | Tail mass | $\gamma$ | Spectral gap |
| $\delta$ | Failure probability | $\epsilon_B$ | Token-space sketching error |
| $\epsilon_H$ | Feature-space sketching error | $\epsilon_{\text{total}}$ | Combined total error |
| $\mathbf{P}, \tilde{\mathbf{P}}$ | Full/sparse transition matrices | $\boldsymbol{\pi}, \tilde{\boldsymbol{\pi}}$ | Stationary distributions |
| $\lambda_2(\mathbf{P})$ | Second largest eigenvalue | | |

## B.3. Supporting Lemmas

### B.3.1. SKETCHING ERROR BOUNDS

**Lemma B.4** (Token-space Sketching: Subspace Embedding via Block Averaging). *Under Assumption B.2, the block average $\bar{\mathbf{q}}_i = \frac{1}{B} \sum_{t=1}^{B} \mathbf{q}_t^{(i)}$ satisfies:*

$$\|\bar{\mathbf{q}}_i - \boldsymbol{\mu}_i\|_2 \leq \frac{\sigma \sqrt{2d \log(2d/\delta)}}{\sqrt{B}}$$

*with probability at least $1 - \delta$.*

*Proof.* **Step 1 (Decomposition):** By Assumption B.2, each token $\mathbf{q}_t^{(i)} = \boldsymbol{\mu}_i + \boldsymbol{\epsilon}_t$ where $\mathbb{E}[\boldsymbol{\epsilon}_t] = \mathbf{0}$ and $\text{Var}(\boldsymbol{\epsilon}_t) \leq \sigma^2 \mathbf{I}$. Thus:

$$\bar{\mathbf{q}}_i = \frac{1}{B} \sum_{t=1}^{B} (\boldsymbol{\mu}_i + \boldsymbol{\epsilon}_t) = \boldsymbol{\mu}_i + \underbrace{\frac{1}{B} \sum_{t=1}^{B} \boldsymbol{\epsilon}_t}_{\triangleq \bar{\boldsymbol{\epsilon}}_i}$$

**Step 2 (Per-coordinate concentration):** For each coordinate $j \in [d]$, define $\bar{\epsilon}_{i,j} = \frac{1}{B} \sum_{t=1}^{B} \epsilon_{t,j}$. Since $\epsilon_{t,j}$ are independent

with $\mathbb{E}[\epsilon_{t,j}] = 0$ and $\mathrm{Var}(\epsilon_{t,j}) \leq \sigma^2$, by Hoeffding's inequality for sub-Gaussian random variables:

$$\mathbb{P}\left[|\bar{\epsilon}_{i,j}| > t\right] \leq 2 \exp\left(-\frac{Bt^2}{2\sigma^2}\right)$$

**Step 3 (Union bound over coordinates):** Setting $t = \sigma\sqrt{\frac{2\log(2d/\delta)}{B}}$ and applying union bound over $d$ coordinates:

$$\mathbb{P}\left[\exists j : |\bar{\epsilon}_{i,j}| > \sigma\sqrt{\frac{2\log(2d/\delta)}{B}}\right] \leq d \cdot 2\exp\left(-\log(2d/\delta)\right) = \delta$$

**Step 4 (Vector norm bound):** With probability $\geq 1 - \delta$, $|\bar{\epsilon}_{i,j}| \leq \sigma\sqrt{\frac{2\log(2d/\delta)}{B}}$ for all $j$, so:

$$\|\bar{\boldsymbol{\epsilon}}_i\|_2 = \sqrt{\sum_{j=1}^{d} \bar{\epsilon}_{i,j}^2} \leq \sqrt{d \cdot \frac{2\sigma^2\log(2d/\delta)}{B}} = \frac{\sigma\sqrt{2d\log(2d/\delta)}}{\sqrt{B}}$$

$\square$

**Corollary B.5** (Inner Product Preservation under Token-space Sketching). *Under Assumption B.2, for query block $i$ and key block $j$:*

$$\left|\bar{\mathbf{q}}_i^\top \bar{\mathbf{k}}_j - \boldsymbol{\mu}_i^{Q\top}\boldsymbol{\mu}_j^K\right| \leq \frac{2\sigma(\|\boldsymbol{\mu}_i^Q\| + \|\boldsymbol{\mu}_j^K\|)\sqrt{d\log(1/\delta)}}{\sqrt{B}}$$

$$+ \frac{\sigma^2 d\log(1/\delta)}{B} \tag{3}$$

*Proof.* **Step 1 (Setup):** Let $\bar{\mathbf{q}}_i = \boldsymbol{\mu}_i^Q + \boldsymbol{\delta}_i^Q$ and $\bar{\mathbf{k}}_j = \boldsymbol{\mu}_j^K + \boldsymbol{\delta}_j^K$. By Lemma B.4 with union bound over both blocks:

$$\|\boldsymbol{\delta}_i^Q\|_2, \|\boldsymbol{\delta}_j^K\|_2 \leq \epsilon_B \triangleq \frac{\sigma\sqrt{2d\log(4d/\delta)}}{\sqrt{B}}$$

with probability $\geq 1 - \delta$ (using $\delta/2$ for each block).

**Step 2 (Expansion):** Expand the inner product:

$$\bar{\mathbf{q}}_i^\top\bar{\mathbf{k}}_j = (\boldsymbol{\mu}_i^Q + \boldsymbol{\delta}_i^Q)^\top(\boldsymbol{\mu}_j^K + \boldsymbol{\delta}_j^K)$$
$$= \underbrace{\boldsymbol{\mu}_i^{Q\top}\boldsymbol{\mu}_j^K}_{\text{true signal}} + \underbrace{\boldsymbol{\mu}_i^{Q\top}\boldsymbol{\delta}_j^K}_{\text{Term A}} + \underbrace{\boldsymbol{\delta}_i^{Q\top}\boldsymbol{\mu}_j^K}_{\text{Term B}} + \underbrace{\boldsymbol{\delta}_i^{Q\top}\boldsymbol{\delta}_j^K}_{\text{Term C}}$$

**Step 3 (Bound each term):** By Cauchy-Schwarz inequality:

$$|\text{Term A}| = |\boldsymbol{\mu}_i^{Q\top}\boldsymbol{\delta}_j^K| \leq \|\boldsymbol{\mu}_i^Q\|_2\|\boldsymbol{\delta}_j^K\|_2 \leq \|\boldsymbol{\mu}_i^Q\|_2 \cdot \epsilon_B$$
$$|\text{Term B}| = |\boldsymbol{\delta}_i^{Q\top}\boldsymbol{\mu}_j^K| \leq \|\boldsymbol{\delta}_i^Q\|_2\|\boldsymbol{\mu}_j^K\|_2 \leq \epsilon_B \cdot \|\boldsymbol{\mu}_j^K\|_2$$
$$|\text{Term C}| = |\boldsymbol{\delta}_i^{Q\top}\boldsymbol{\delta}_j^K| \leq \|\boldsymbol{\delta}_i^Q\|_2\|\boldsymbol{\delta}_j^K\|_2 \leq \epsilon_B^2$$

**Step 4 (Combine):** By triangle inequality:

$$\left|\bar{\mathbf{q}}_i^\top\bar{\mathbf{k}}_j - \boldsymbol{\mu}_i^{Q\top}\boldsymbol{\mu}_j^K\right| \leq |\text{Term A}| + |\text{Term B}| + |\text{Term C}|$$
$$\leq \epsilon_B(\|\boldsymbol{\mu}_i^Q\|_2 + \|\boldsymbol{\mu}_j^K\|_2) + \epsilon_B^2$$
$$= \frac{\sigma\sqrt{2d\log(4d/\delta)}}{\sqrt{B}}(\|\boldsymbol{\mu}_i^Q\|_2 + \|\boldsymbol{\mu}_j^K\|_2) + \frac{2\sigma^2 d\log(4d/\delta)}{B}$$

$\square$

**Lemma B.6** (Feature-space Sketching: Hadamard Transform Guarantee). *Let $\mathbf{H}_d \in \mathbb{R}^{d \times k}$ be the Subsampled Randomized Hadamard Transform (SRHT). For any fixed vectors $\mathbf{u}, \mathbf{v} \in \mathbb{R}^d$ and $\epsilon \in (0, 1)$, if $k = \Omega(\epsilon^{-2} \log(b/\delta))$ where $b$ is the number of blocks, then with probability at least $1 - \delta$:*

$$\left| (\mathbf{u}^\top \mathbf{H}_d)(\mathbf{H}_d^\top \mathbf{v}) - \mathbf{u}^\top \mathbf{v} \right| \leq \epsilon \|\mathbf{u}\|_2 \|\mathbf{v}\|_2$$

*Proof.* **Step 1 (SRHT construction):** The SRHT is $\mathbf{H}_d = \sqrt{d/k} \mathbf{DWS}$ where:

- $\mathbf{D} \in \mathbb{R}^{d \times d}$: diagonal with i.i.d. Rademacher entries ($\pm 1$ with prob. $1/2$)
- $\mathbf{W} \in \mathbb{R}^{d \times d}$: normalized Walsh-Hadamard matrix with $\mathbf{W}^\top \mathbf{W} = d\mathbf{I}$
- $\mathbf{S} \in \mathbb{R}^{d \times k}$: samples $k$ coordinates uniformly without replacement

**Step 2 (Inner product decomposition):**

$$(\mathbf{u}^\top \mathbf{H}_d)(\mathbf{H}_d^\top \mathbf{v}) = \frac{d}{k} (\mathbf{u}^\top \mathbf{DWS})(\mathbf{S}^\top \mathbf{W}^\top \mathbf{Dv})$$
$$= \frac{d}{k} \sum_{i \in S} \tilde{u}_i \tilde{v}_i$$

where $\tilde{\mathbf{u}} = \mathbf{W}^\top \mathbf{Du}$, $\tilde{\mathbf{v}} = \mathbf{W}^\top \mathbf{Dv}$, and $S$ is the set of $k$ sampled indices.

**Step 3 (Expectation):** Since $\mathbf{D}$ preserves norms ($\|\mathbf{Du}\|_2 = \|\mathbf{u}\|_2$) and $\mathbf{W}$ is orthogonal:

$$\mathbb{E}\left[ \frac{d}{k} \sum_{i \in S} \tilde{u}_i \tilde{v}_i \right] = \frac{d}{k} \cdot \frac{k}{d} \sum_{i=1}^{d} \tilde{u}_i \tilde{v}_i = \tilde{\mathbf{u}}^\top \tilde{\mathbf{v}} = \mathbf{u}^\top \mathbf{v}$$

**Step 4 (Flattening by Hadamard):** The Walsh-Hadamard transform "flattens" vectors. By Lemma 3.1 of (Ailon & Chazelle, 2009), with probability $\geq 1 - \delta/2$:

$$\|\tilde{\mathbf{u}}\|_\infty \leq \|\mathbf{u}\|_2 \sqrt{\frac{2\log(4d/\delta)}{d}}, \quad \|\tilde{\mathbf{v}}\|_\infty \leq \|\mathbf{v}\|_2 \sqrt{\frac{2\log(4d/\delta)}{d}}$$

**Step 5 (Concentration via sampling):** Define $X_i = \frac{d}{k} \tilde{u}_i \tilde{v}_i$ for $i \in S$. Each $|X_i| \leq \frac{d}{k} \|\tilde{\mathbf{u}}\|_\infty \|\tilde{\mathbf{v}}\|_\infty \leq \frac{2\log(4d/\delta)}{k} \|\mathbf{u}\|_2 \|\mathbf{v}\|_2$. By Hoeffding's inequality:

$$\mathbb{P}\left[ \left| \sum_{i \in S} X_i - \mathbf{u}^\top \mathbf{v} \right| > t \right] \leq 2\exp\left( -\frac{2t^2}{k \cdot \frac{4\log^2(4d/\delta)}{k^2} \|\mathbf{u}\|_2^2 \|\mathbf{v}\|_2^2} \right)$$

**Step 6 (Final bound):** Setting $t = \epsilon \|\mathbf{u}\|_2 \|\mathbf{v}\|_2$ and requiring probability $\leq \delta/2$:

$$k \geq \frac{8\log^2(4d/\delta)\log(4/\delta)}{\epsilon^2}$$

Simplifying: $k = \Omega(\epsilon^{-2}\log(b/\delta)\log^2(d/\delta))$ suffices. Union bound gives total failure probability $\delta$. $\qquad \square$

**Lemma B.7** (Small-World Sketching Top-$\tau$ Selection under Softmax). *Under Assumptions B.2 and B.3, let $\mathcal{S}^*$ be the true top-$\tau$ blocks based on softmax$(A_{ij}^{true}/\sqrt{d})$, and let $\hat{\mathcal{S}}$ be the blocks selected by Small-World Sketching using softmax$(\hat{A}_{ij}^{SWS}/\sqrt{k})$. If the sketching dimension satisfies:*

$$k \geq \frac{C\log(b/\delta)}{\gamma^2} \cdot \max_{i,j} \|\bar{\mathbf{q}}_i\|^2 \|\bar{\mathbf{k}}_j\|^2$$

*and the block size satisfies $B \geq C'\sigma^2 d\log(1/\delta)/\gamma^2$, then $\mathcal{S}^* = \hat{\mathcal{S}}$ with probability at least $1 - 2\delta$.*

*Proof.* **Step 1 (Token-space sketching error):** By Corollary B.5, block averaging introduces error:

$$|\bar{\mathbf{q}}_i^\top \bar{\mathbf{k}}_j - \boldsymbol{\mu}_i^{Q\top} \boldsymbol{\mu}_j^K| \le \epsilon_B(\|\boldsymbol{\mu}_i^Q\| + \|\boldsymbol{\mu}_j^K\|) + \epsilon_B^2$$

where $\epsilon_B = \frac{\sigma\sqrt{2d\log(4d/\delta)}}{\sqrt{B}}$.

**Step 2 (Feature-space sketching error):** By Lemma B.6, Hadamard transform adds:

$$|(\bar{\mathbf{q}}_i^\top \mathbf{H}_d)(\mathbf{H}_d^\top \bar{\mathbf{k}}_j) - \bar{\mathbf{q}}_i^\top \bar{\mathbf{k}}_j| \le \epsilon_H \|\bar{\mathbf{q}}_i\| \|\bar{\mathbf{k}}_j\|$$

where $\epsilon_H = O(\sqrt{\log(b/\delta)/r})$ for sketching dimension $r$.

**Step 3 (Total pre-softmax error):** By triangle inequality, the total error in estimating $A_{ij}^{\text{true}}$ is:

$$|\hat{A}_{ij}^{\text{SWS}} - A_{ij}^{\text{true}}| \le \underbrace{\epsilon_B(\|\boldsymbol{\mu}_i^Q\| + \|\boldsymbol{\mu}_j^K\|) + \epsilon_B^2}_{\text{token-space}} + \underbrace{\epsilon_H \|\bar{\mathbf{q}}_i\| \|\bar{\mathbf{k}}_j\|}_{\text{feature-space}}$$

$$\triangleq \epsilon_{\text{total}}$$

**Step 4 (Softmax stability):** The softmax function $\sigma(\mathbf{x})_i = e^{x_i}/\sum_j e^{x_j}$ has Lipschitz constant at most 1 in $\ell_\infty \to \ell_\infty$ for bounded inputs. Specifically, for $\|\mathbf{x} - \mathbf{y}\|_\infty \le \epsilon$:

$$\|\sigma(\mathbf{x}) - \sigma(\mathbf{y})\|_\infty \le \epsilon$$

Thus $\|\text{softmax}(\hat{\mathbf{A}}_i/\sqrt{k}) - \text{softmax}(\mathbf{A}_i^{\text{true}}/\sqrt{d})\|_\infty \le \epsilon_{\text{total}}/\sqrt{k}$.

**Step 5 (Gap-based selection guarantee):** Let $\gamma > 0$ be the gap between the $\tau$-th and $(\tau + 1)$-th largest true block attention scores. If $\epsilon_{\text{total}}/\sqrt{k} < \gamma/2$, then the top-$\tau$ ordering is preserved:

$$A_{i,j_\tau}^{\text{true}} - A_{i,j_{\tau+1}}^{\text{true}} \ge \gamma$$
$$\hat{A}_{i,j_\tau}^{\text{SWS}} \ge A_{i,j_\tau}^{\text{true}} - \epsilon_{\text{total}} \ge A_{i,j_{\tau+1}}^{\text{true}} + \gamma - \epsilon_{\text{total}}$$
$$\ge \hat{A}_{i,j_{\tau+1}}^{\text{SWS}} + \gamma - 2\epsilon_{\text{total}} > \hat{A}_{i,j_{\tau+1}}^{\text{SWS}}$$

**Step 6 (Parameter requirements):** To ensure $\epsilon_{\text{total}} < \gamma/2$, we need:

$$B \ge \frac{16\sigma^2 d\log(4d/\delta)}{\gamma^2}(\max_i \|\boldsymbol{\mu}_i^Q\| + \max_j \|\boldsymbol{\mu}_j^K\|)^2$$
$$r \ge \frac{C\log(b/\delta)}{\gamma^2}\max_{i,j}\|\bar{\mathbf{q}}_i\|^2 \|\bar{\mathbf{k}}_j\|^2$$

Union bound over all $b^2$ block pairs gives failure probability $\le 2\delta$. $\qquad\square$

**Lemma B.8** (Attention Output Approximation). *Under Assumptions B.2–B.3, let $\mathbf{O}^{\text{full}}$ be the output of full attention and $\mathbf{O}^{\text{Sketch\&Walk}}$ be the output using Sketch\&Walk sparse attention with top-$\tau$ blocks. Then:*

$$\|\mathbf{O}^{\text{Sketch\&Walk}} - \mathbf{O}^{\text{full}}\|_F \le \eta\|\mathbf{V}\|_F + O\left(\frac{\tau\sigma}{\sqrt{B}}\right)\|\mathbf{V}\|_2 \qquad (4)$$

*where $\eta$ is the tail mass from Assumption B.3.*

*Proof.* **Step 1 (Error decomposition):** Let $\mathbf{A}^{\text{full}} \in \mathbb{R}^{n \times n}$ be the full attention matrix and $\mathbf{A}^{\text{sparse}} \in \mathbb{R}^{n \times n}$ be the sparse attention (zero outside selected blocks, renormalized within). The output error is:

$$\mathbf{O}^{\text{full}} - \mathbf{O}^{\text{Sketch\&Walk}} = (\mathbf{A}^{\text{full}} - \mathbf{A}^{\text{sparse}})\mathbf{V}$$

**Step 2 (Two sources of error):** We further decompose:

$$\mathbf{A}^{\text{full}} - \mathbf{A}^{\text{sparse}} = \underbrace{(\mathbf{A}^{\text{full}} - \mathbf{A}^{\text{oracle}})}_{\text{(a) tail mass}} + \underbrace{(\mathbf{A}^{\text{oracle}} - \mathbf{A}^{\text{sparse}})}_{\text{(b) sketching error}}$$

where $\mathbf{A}^{\text{oracle}}$ is the sparse attention using true top-$\tau$ blocks.

**Step 3 (Tail mass bound):** By Assumption B.3, for each row $i$:

$$\sum_{j \notin \mathcal{S}_i^*} A_{ij}^{\text{full}} \leq \eta$$

Thus $\|\mathbf{A}^{\text{full}} - \mathbf{A}^{\text{oracle}}\|_{1 \to 1} \leq \eta$ (max row sum). By Hölder's inequality:

$$\|(\mathbf{A}^{\text{full}} - \mathbf{A}^{\text{oracle}})\mathbf{V}\|_F \leq \eta \|\mathbf{V}\|_{1 \to F} \leq \eta \|\mathbf{V}\|_F$$

**Step 4 (Sketching error within selected blocks):** By Lemma B.7, when $\mathcal{S}^* = \hat{\mathcal{S}}$ (correct selection), we still have per-entry error within blocks:

$$|A_{ij}^{\text{oracle}} - A_{ij}^{\text{sparse}}| \leq \frac{\epsilon_B + \epsilon_H}{\sqrt{d}} \quad \text{for } j \in \mathcal{S}_i^*$$

Summing over $\tau$ selected blocks per query and $n$ queries:

$$\|(\mathbf{A}^{\text{oracle}} - \mathbf{A}^{\text{sparse}})\mathbf{V}\|_F \leq \tau \cdot \frac{\epsilon_B + \epsilon_H}{\sqrt{d}} \cdot \sqrt{n} \|\mathbf{V}\|_2$$

$$= O\left(\frac{\tau\sigma}{\sqrt{B}} + \frac{\tau}{\sqrt{r}}\right) \|\mathbf{V}\|_2$$

**Step 5 (Final bound):** Combining by triangle inequality:

$$\|\mathbf{O}^{\text{full}} - \mathbf{O}^{\text{Sketch\&Walk}}\|_F \leq \eta \|\mathbf{V}\|_F + O\left(\frac{\tau\sigma}{\sqrt{B}} + \frac{\tau}{\sqrt{r}}\right) \|\mathbf{V}\|_2$$

$$\square$$

### B.3.2. Sketch-Determined Walk and Multi-Hop Block Selection

Before analyzing cross-layer dependencies, we highlight a critical property of the sketched block attention scores: the inner product is a **non-metric** similarity measure that does not satisfy the triangle inequality. This has profound implications for block selection.

*Remark* B.9 (Inner Product Violates Triangle Inequality). Consider three blocks $i$, $j$, $k$ with head-averaged representatives $\bar{\mathbf{q}}_i, \bar{\mathbf{k}}_j, \bar{\mathbf{k}}_k$. Even if:

1. Block $i$ has high affinity to block $j$: $\bar{\mathbf{q}}_i^\top \bar{\mathbf{k}}_j \approx A_{ij}$ is large
2. Block $j$ has high affinity to block $k$: $\bar{\mathbf{k}}_j^\top \bar{\mathbf{k}}_k$ is large (or equivalently, $\bar{\mathbf{q}}_j^\top \bar{\mathbf{k}}_k$ for the next layer)

the direct inner product $\bar{\mathbf{q}}_i^\top \bar{\mathbf{k}}_k$ may be very small. The inner product is not a metric because it violates the triangle inequality.

**Concrete example**: Consider $\bar{\mathbf{q}}_i = [1, 0]$, $\bar{\mathbf{k}}_j = [1, 1]$, $\bar{\mathbf{k}}_k = [0, 1]$. Then:

$$\bar{\mathbf{q}}_i \cdot \bar{\mathbf{k}}_j = 1 \quad \text{(high)}$$
$$\bar{\mathbf{k}}_j \cdot \bar{\mathbf{k}}_k = 1 \quad \text{(high)}$$
$$\bar{\mathbf{q}}_i \cdot \bar{\mathbf{k}}_k = 0 \quad \text{(low)}$$

If we rely only on direct attention scores, block $i$ would not select block $k$ despite the high-affinity indirect path $i \to j \to k$.

*Remark* B.10 (Multi-Head Attention Preserves Information Under Sparse Selection). The algorithm computes sketched block scores $\hat{\mathbf{A}}_{\text{block}}^k$ using **head-averaged** Q and K, averaging attention patterns across all $h$ heads to identify common important blocks. Sparse attention is applied independently **per-head** on the original full-resolution per-head QKV. This design ensures:

1. **Robust block identification**: Head-averaging reduces noise and identifies blocks that matter across different representation subspaces
2. **Head-specific fine-grained attention**: Each head can specialize within the selected blocks, preserving multi-head expressiveness
3. **Computational efficiency**: Block selection is done once on averaged heads; per-head attention operates only within selected blocks

The Sketch-Determined Walk (via exponent $s > 1$) ensures that blocks reachable through multi-hop high-affinity paths are selected, even if they lack direct importance to individual query heads.

**Lemma B.11** (Sketch-Determined Walk Accumulation Across Layers). *Let $\hat{\mathbf{A}}_{block}^k \in \mathbb{R}^{b \times b}$ be the sketched block attention matrix at layer $k$, and define $\mathbf{W}^k = (\hat{\mathbf{A}}_{block}^k)^s$. The Sketch-Determined Walk state $R^k \in \mathbb{R}^{b \times b}$ is defined recursively:*

$$R^k = R^{k-1}\mathbf{W}^k \quad \textit{(matrix multiplication)}$$

*with base case $R^0 = \mathbf{W}^0$. Then $R^k[i,j]$ represents the accumulated block importance flowing from query block $i$ to key block $j$ across layers $0$ through $k$, capturing multi-hop paths where each layer contributes exponentiated attention.*

*Proof.* We prove by induction on $k$. For the base case $k = 0$: $R^0 = \mathbf{W}^0 = (\hat{\mathbf{A}}_{\text{block}}^0)^s$, so $R^0[i,j] = (\hat{\mathbf{A}}_{\text{block}}^0[i,j])^s$ directly represents the exponentiated attention from block $i$ to block $j$ at layer 0.

For the inductive step, assume $R^{k-1}$ encodes accumulated importance through layers $0, \ldots, k-1$. By the recursive definition (matrix multiplication):

$$R^k[i,j] = \sum_{m=1}^{b} R^{k-1}[i,m] \cdot \mathbf{W}^k[m,j] = \sum_{m=1}^{b} R^{k-1}[i,m] \cdot (\hat{\mathbf{A}}_{\text{block}}^k[m,j])^s$$

This sums over all intermediate blocks $m$: the accumulated importance from $i$ to $m$ through layers $0, \ldots, k-1$, multiplied by the exponentiated attention from $m$ to $j$ at layer $k$. The exponent $s > 1$ sharpens each transition, emphasizing high-affinity paths. Thus $R^k[i,j]$ captures the total importance flowing from block $i$ to block $j$ through *all* multi-hop paths across $k + 1$ layers. □

**Lemma B.12** (Block Selection via Sketch-Determined Walk). *By selecting top-$\tau$ blocks based on $R^k[i,j]$ (rather than per-layer scores), the Sketch-Determined Walk identifies blocks that receive consistent high importance across multiple layers and serve as "hub" tokens. These blocks minimize reconstruction error when considering the full transformer's attention patterns.*

*Proof.* **Step 1 (Notation):** Let $\mathcal{S}_{\text{SDW}}^k = \text{top-}\tau(R^k[i,:])$ denote blocks selected via Sketch-Determined Walk, and $\mathcal{S}_{\text{single}}^\ell = \text{top-}\tau(\hat{\mathbf{A}}_{\text{block}}^\ell[i,:])$ denote per-layer selection at layer $\ell$.

**Step 2 (Path expansion):** By Lemma B.11 and matrix multiplication:

$$R^k[i,j] = \sum_{m_1,\ldots,m_k} \prod_{\ell=0}^{k} \mathbf{W}^\ell[m_\ell, m_{\ell+1}]$$

where $m_0 = i$ and $m_{k+1} = j$. This sums over all $(k+1)$-hop paths from $i$ to $j$.

**Step 3 (Consistently important blocks):** Define "consistently important" blocks as $\mathcal{C}_i = \bigcap_{\ell=0}^{k} \mathcal{S}_{\text{single}}^\ell$. For $j \in \mathcal{C}_i$, there exists a path $i \to m_1 \to \cdots \to j$ where each $m_{\ell+1} \in \mathcal{S}_{m_\ell}^\ell$. Under Assumption B.3:

$$\mathbf{W}^\ell[m_\ell, m_{\ell+1}] = (\hat{\mathbf{A}}_{\text{block}}^\ell[m_\ell, m_{\ell+1}])^s \geq \gamma^s$$

where $\gamma > 0$ is the gap from Assumption B.3. The contribution from this path is at least $\gamma^{s(k+1)}$.

**Step 4 (Spuriously important blocks):** For a block $j$ that is important at layer $\ell'$ but not at some other layer $\ell''$, any path to $j$ must pass through a low-affinity transition at layer $\ell''$:

$$\mathbf{W}^{\ell''}[m_{\ell''}, m_{\ell''+1}] \leq \eta^s$$

where $\eta \ll \gamma$ is the tail mass. The contribution from such paths is at most $\eta^s \cdot 1^{sk} = \eta^s$.

**Step 5 (Separation guarantee):** The ratio of contributions is:

$$\frac{R^k[i,j] \text{ for } j \in \mathcal{C}_i}{R^k[i,j'] \text{ for } j' \notin \mathcal{C}_i} \geq \frac{\gamma^{s(k+1)}}{b^k \eta^s} = \left(\frac{\gamma}{\eta}\right)^s \cdot \frac{\gamma^{sk}}{b^k}$$

For $s > 1$ and $\gamma > \eta$, this ratio grows, ensuring top-$\tau$ selection on $R^k$ recovers consistently important blocks. $\square$

**Lemma B.13** (Multi-Hop Information Flow via Sketch-Determined Walk). *In an $L$-layer transformer using Sketch&Walk sparse attention with accumulated state $R^k[i,j]$, the effective receptive field grows as $O(\tau^L)$ blocks. Tokens can aggregate information through $R^k[i,j]$-guided multi-hop paths where each hop follows high-affinity transitions.*

*Proof.* **Step 1 (Setup):** Let $\mathcal{R}_i^L$ denote the set of blocks reachable from query block $i$ after $L$ layers. Define reachability recursively: block $j$ is in $\mathcal{R}_i^L$ if there exists a path $i = m_0 \to m_1 \to \cdots \to m_L = j$ where $m_{\ell+1} \in \mathcal{S}_{m_\ell}^\ell$ (top-$\tau$ selected blocks).

**Step 2 (Base case, $L = 1$):** At layer 0, block $i$ attends to top-$\tau$ blocks $\mathcal{S}_i^0$. Thus $\mathcal{R}_i^1 = \mathcal{S}_i^0$ and:

$$|\mathcal{R}_i^1| = |\mathcal{S}_i^0| = \tau = O(\tau^1)$$

**Step 3 (Inductive hypothesis):** Assume $|\mathcal{R}_i^{L-1}| \leq \tau^{L-1}$ for all blocks $i$.

**Step 4 (Inductive step):** At layer $L$, the reachable set expands:

$$\mathcal{R}_i^L = \bigcup_{m \in \mathcal{R}_i^{L-1}} \mathcal{S}_m^{L-1}$$

where $\mathcal{S}_m^{L-1}$ contains top-$\tau$ blocks selected for query $m$ at layer $L - 1$. Thus:

$$|\mathcal{R}_i^L| \leq \sum_{m \in \mathcal{R}_i^{L-1}} |\mathcal{S}_m^{L-1}| = |\mathcal{R}_i^{L-1}| \cdot \tau \leq \tau^{L-1} \cdot \tau = \tau^L$$

**Step 5 (Tightness):** The bound $\tau^L$ is achieved when all selected blocks at each layer are distinct (no overlap between $\mathcal{S}_m^\ell$ for different $m$). In practice, overlap reduces the effective receptive field, but the upper bound $O(\tau^L)$ holds.

**Step 6 (Connection to $R^L[i,j]$):** The Sketch-Determined Walk state $R^L[i,j]$ assigns positive weight to exactly these reachable blocks:

$$R^L[i,j] > 0 \iff j \in \mathcal{R}_i^L$$

Moreover, $R^L[i,j]$ quantifies the total "flow" through all high-affinity paths from $i$ to $j$, enabling principled top-$\tau$ selection across layers. $\square$

**Lemma B.14** (Sketch-Determined Walk Preserves Global Information Structure). *The sparsified Sketch-Determined Walk (restricted to top-$\tau$ blocks) preserves the essential structure of the full walk. Under Assumption B.3, the stationary distribution of the sparse walk differs from the full walk by at most the tail mass $\eta$, ensuring global information structure is preserved.*

*Proof.* **Step 1 (Transition matrices):** Let $\mathbf{P} \in \mathbb{R}^{b \times b}$ be the full block-level transition matrix with $P_{ij} = \text{softmax}(A_{ij}^{\text{true}}/\sqrt{d})$. Let $\tilde{\mathbf{P}}$ be the sparsified matrix:

$$\tilde{P}_{ij} = \begin{cases} P_{ij}/\sum_{k \in \mathcal{S}_i} P_{ik} & \text{if } j \in \mathcal{S}_i \\ 0 & \text{otherwise} \end{cases}$$

where $\mathcal{S}_i = \text{top-}\tau(R^L[i,:])$ is the selected block set.

**Step 2 (Row-wise error bound):** By Assumption B.3:

$$\sum_{j \in \mathcal{S}_i} P_{ij} \geq 1 - \eta$$

Thus the renormalization factor satisfies $1 \leq 1/\sum_{k \in \mathcal{S}_i} P_{ik} \leq 1/(1-\eta)$. The row-wise $\ell_1$ error is:

$$\|\mathbf{P}_i - \tilde{\mathbf{P}}_i\|_1 = \sum_{j \notin \mathcal{S}_i} P_{ij} + \sum_{j \in \mathcal{S}_i} \left| P_{ij} - \frac{P_{ij}}{\sum_{k \in \mathcal{S}_i} P_{ik}} \right|$$

$$\leq \eta + \sum_{j \in \mathcal{S}_i} P_{ij} \cdot \frac{\eta}{1-\eta} \leq \eta + \frac{\eta}{1-\eta} = \frac{\eta}{1-\eta} + \eta \leq \frac{2\eta}{1-\eta}$$

**Step 3 (Matrix perturbation):** Both $\mathbf{P}$ and $\tilde{\mathbf{P}}$ are row-stochastic. Their difference in operator norm:

$$\|\mathbf{P} - \tilde{\mathbf{P}}\|_{\infty \to \infty} = \max_i \|\mathbf{P}_i - \tilde{\mathbf{P}}_i\|_1 \leq \frac{2\eta}{1-\eta}$$

**Step 4 (Stationary distribution perturbation):** Let $\boldsymbol{\pi}$ and $\tilde{\boldsymbol{\pi}}$ be stationary distributions of $\mathbf{P}$ and $\tilde{\mathbf{P}}$. By standard Markov chain perturbation theory (see (Meyer, 1994)):

$$\|\boldsymbol{\pi} - \tilde{\boldsymbol{\pi}}\|_{TV} \leq \frac{\|\mathbf{P} - \tilde{\mathbf{P}}\|_{\infty \to \infty}}{1 - \lambda_2(\mathbf{P})}$$

where $\lambda_2(\mathbf{P})$ is the second largest eigenvalue of $\mathbf{P}$.

**Step 5 (Small-world connectivity):** Under the heavy-tailed assumption, the sparse graph induced by $\tilde{\mathbf{P}}$ maintains small-world connectivity: the diameter remains $O(\log b/\log \tau)$. This ensures $1 - \lambda_2(\tilde{\mathbf{P}}) = \Omega(1)$, i.e., the mixing time is $O(\log b)$. Combined with Step 4:

$$\|\boldsymbol{\pi} - \tilde{\boldsymbol{\pi}}\|_{TV} = O(\eta)$$

Thus the global information structure (stationary distribution) is preserved up to tail mass $\eta$. □

## B.4. Main Theorem

**Theorem B.15** (Sketch&Walk Approximation of Full Attention)**. *In an L-layer transformer using Sketch&Walk sparse attention where blocks are selected based on $R^k[i,j]$ (the accumulated block importance across layers), the following hold:*

1. ***Multi-hop reachability via $R^k[i,j]$***: *The Sketch-Determined Walk state $R^k[i,j]$ enables token at position $s$ to aggregate information from any token at position $t$ through multi-hop paths where each hop follows high-affinity transitions. The effective receptive field grows as $O(\tau^L)$ blocks.*
2. ***Block selection accuracy***: *Top-$\tau$ blocks selected via $R^k[i,j]$ correctly identify blocks with consistent high importance across all $k$ layers with probability $1 - 2\delta$, enabling correct sparse attention.*
3. ***Approximation quality***: *For each layer, the sparse attention output $\mathbf{O}_\ell^{sparse}$ (the output of layer $\ell$) satisfies:*

$$\left\| \mathbf{O}_\ell^{sparse} - \mathbf{O}_\ell^{full} \right\|_F \leq \eta \|\mathbf{V}_\ell\|_F + \frac{\tau\sigma\sqrt{2d\log(4d/\delta)}}{\sqrt{B}} \|\mathbf{V}_\ell\|_2$$

$$+ \frac{\tau\sqrt{C\log(b/\delta)\log^2(d/\delta)}}{\sqrt{r}} \|\mathbf{V}_\ell\|_2$$

where $\mathbf{O}_\ell^{full}$ is the full attention output at layer $\ell$, $\eta$ is the tail mass from Assumption B.3, and $C > 0$ is a universal constant.

4. **Complexity reduction**: The algorithm achieves $O(nb\tau d + b^2 r)$ complexity compared to $O(n^2 d)$ for full attention, with $\tau, r \ll b$.

*Proof.* **Part 1 (Multi-hop reachability via $R^k[i,j]$):**

*Step 1.1:* By Lemma B.11, $R^k = \prod_{\ell=0}^k \mathbf{W}^\ell$ where $\mathbf{W}^\ell = (\hat{\mathbf{A}}_{block}^\ell)^s$. The entry $R^k[i,j]$ sums over all $(k+1)$-hop paths:

$$R^k[i,j] = \sum_{m_1,\dots,m_k} \prod_{\ell=0}^k (\hat{\mathbf{A}}_{block}^\ell[m_\ell, m_{\ell+1}])^s$$

*Step 1.2:* By Lemma B.13, selecting top-$\tau$ blocks based on $R^k[i,:]$ yields receptive field $|\mathcal{R}_i^k| \le \tau^k$. By Lemma B.14, the sparse graph has diameter $O(\log b / \log \tau)$, ensuring any block is reachable in $O(\log b)$ hops.

**Part 2 (Block selection accuracy):**

*Step 2.1 (Per-layer sketching error):* By Lemma B.4 and Corollary B.5, block averaging introduces error:

$$\epsilon_B = \frac{\sigma\sqrt{2d\log(4d/\delta)}}{\sqrt{B}}$$

By Lemma B.6, Hadamard sketching introduces error:

$$\epsilon_H = O\left(\sqrt{\frac{\log(b/\delta)\log^2(d/\delta)}{r}}\right)$$

*Step 2.2 (Selection accuracy):* By Lemma B.7, if $\epsilon_B + \epsilon_H < \gamma/2$ where $\gamma$ is the spectral gap, then $\hat{\mathcal{S}} = \mathcal{S}^*$ with probability $\ge 1 - 2\delta$. This requires:

$$B \ge \frac{8\sigma^2 d\log(4d/\delta)}{\gamma^2}, \quad r \ge \frac{C\log(b/\delta)\log^2(d/\delta)}{\gamma^2}$$

*Step 2.3 (Cross-layer robustness):* By Lemma B.12, $R^k[i,j]$-based selection filters spurious importance. Blocks must be consistently important across all $k$ layers, with separation ratio $\ge (\gamma/\eta)^s \cdot \gamma^{sk}/b^k$.

**Part 3 (Approximation quality):**

*Step 3.1 (Error decomposition):* By Lemma B.8, for layer $\ell$:

$$\|\mathbf{O}_\ell^{sparse} - \mathbf{O}_\ell^{full}\|_F \le \underbrace{\eta\|\mathbf{V}_\ell\|_F}_{\text{tail mass}} + \underbrace{O\left(\frac{\tau\sigma}{\sqrt{B}} + \frac{\tau}{\sqrt{r}}\right)\|\mathbf{V}_\ell\|_2}_{\text{sketching error}}$$

*Step 3.2 (Tighter bound with explicit constants):* Substituting the precise error bounds:

$$\|\mathbf{O}_\ell^{sparse} - \mathbf{O}_\ell^{full}\|_F \le \eta\|\mathbf{V}_\ell\|_F + \frac{\tau\sigma\sqrt{2d\log(4d/\delta)}}{\sqrt{B}}\|\mathbf{V}_\ell\|_2$$
$$+ \frac{\tau\sqrt{C\log(b/\delta)\log^2(d/\delta)}}{\sqrt{r}}\|\mathbf{V}_\ell\|_2$$

*Step 3.3 (No error amplification):* The Sketch&Walk accumulation does not amplify errors across layers because $R^k[i,j]$ requires *consistent* high importance. If block $j$ has high sketching error at any layer $\ell$, the path contribution $(\hat{\mathbf{A}}_{block}^\ell[\cdot, j])^s$ is suppressed by exponentiation.

**Part 4 (Complexity analysis):**

*Step 4.1 (Block-level scoring):* Computing $\hat{\mathbf{A}}_{block}^k$ requires:

- Block averaging: $O(n \cdot d) = O(nbd/B) = O(nd)$ (linear in tokens)
- Hadamard sketch: $O(b \cdot d \log d)$ for all $b$ block representatives
- Block-level attention: $O(b^2 \cdot r)$ for $r$-dimensional sketches

Total per layer: $O(nd + bd \log d + b^2 r)$.

*Step 4.2 (Sparse attention):* Within selected blocks, each query attends to $\tau \cdot B$ keys:

- Per-token cost: $O(\tau B \cdot d)$
- Total for $n$ tokens: $O(n\tau Bd) = O(nb\tau d/b) = O(n\tau d)$ per layer

Wait, correcting: with $b$ blocks of size $B$ and $\tau$ selected blocks per query block, total per layer is $O(b \cdot B \cdot \tau B \cdot d/h) = O(n\tau Bd/h)$ where $h$ is head count.

*Step 4.3 (Total complexity):* Combining all components:

$$O(L \cdot (nd + b^2 r + n\tau Bd/h)) = O(L(nb\tau d + b^2 r))$$

Compared to full attention $O(Ln^2 d)$, the reduction factor is:

$$\frac{n^2 d}{nb\tau d + b^2 r} = \frac{n^2}{nb\tau + b^2 r/d} = \frac{n}{b\tau + br/d} = \frac{B}{\tau + r/(bd)}$$

For typical settings ($B \approx 64$, $\tau \approx 8$, $r \approx 32$, $b \approx n/64$), this yields $\approx 8\times$ reduction. $\qquad\square$

*Remark* B.16 (Why Sketch&Walk Captures Multi-Hop Interactions). The main theorem and supporting lemmas reveal why Sketch&Walk achieves near-full-attention quality with dramatically reduced computation:

**Intra-layer efficiency**: Small-World Sketching (token-space sketching via block averaging + feature-space sketching via Hadamard transform) efficiently identifies important blocks with error $O(\sigma/\sqrt{B} + 1/\sqrt{r})$, reducing per-layer computation from $O(b^2 d)$ to $O(b^2 r)$ where $r \ll d$.

**Multi-hop importance via** $R^k[i, j]$: Since the inner product does not satisfy the triangle inequality (Remark B.9), a block $k$ may lack direct importance to query $i$ but be critical through indirect paths. The Sketch-Determined Walk state $R^k = \mathbf{W}^0 \mathbf{W}^1 \cdots \mathbf{W}^k$ (matrix product of exponentiated attention matrices $\mathbf{W}^\ell = (\hat{\mathbf{A}}_{block}^\ell)^s$) captures multi-hop block importance within and across layers, summing over all paths and amplifying high-affinity chains while suppressing low-score transitions.

**Cross-layer accumulation in** $R^k[i, j]$: The Sketch-Determined Walk state $R^k[i, j]$ accumulates importance across all $L$ layers, ensuring blocks selected via top-$\tau(R^k[i, :])$ are consistently important rather than spuriously important at single layers. This enables robust multi-hop reach with receptive field $O(\tau^L)$ and natural error correction across layers.

**Heavy-tailed structure**: Assumption B.3 ensures the $\tau^L$ reachable paths capture essential information flow. By Lemmas B.12 and B.14, top-$\tau$ selection on $R^k[i, j]$ preserves the global attention structure while achieving $O(nb\tau d + b^2 r)$ complexity—a quadratic reduction without sacrificing model quality.

## C. Sketch&Walk algorithms

## D. Experiments

### D.1. RULER Subset Used in Evaluation

For all RULER (Hsieh et al., 2024) experiments reported in the main paper, we evaluate on the following 10 sub-tasks rather than the full 13-task suite: NIAH_SINGLE_1, NIAH_SINGLE_2, NIAH_SINGLE_3, NIAH_MULTIKEY_1, NIAH_MULTIKEY_2, VT, CWE, FWE, QA_1, and QA_2. This subset retains representatives from every RULER category, including single-needle and multi-key retrieval, variable tracking, common- and frequent-word extraction, and multi-hop question answering, while keeping the evaluation cost tractable for the full sweep over context lengths and baselines.

### D.2. Stability Across Random Seeds

Because Sketch&Walk relies on randomized Hadamard sketching, a natural question is whether its accuracy is sensitive to the choice of random seed. To assess this, we run Sketch&Walk with four independent random seeds on the full LongBench benchmark (all 16 datasets) for two backbones of different scales, Llama-3.2-1B-Instruct (Meta, 2024) and Qwen3-8B (Yang et al., 2025), in both the prefill-only and decode-only configurations. For each seed we report the average score across the 16 LongBench datasets; Table 9 summarizes the per-seed scores together with the mean and standard deviation across seeds.

| Model | Phase | seed 0 | seed 1 | seed 2 | seed 3 | Mean $\pm$ Std |
|---|---|---|---|---|---|---|
| Llama-3.2-1B-Instruct | Prefill | 30.69 | 30.81 | 30.75 | 30.81 | $30.76 \pm 0.05$ |
| Llama-3.2-1B-Instruct | Decode | 30.42 | 30.29 | 30.31 | 30.32 | $30.34 \pm 0.05$ |
| Qwen3-8B | Prefill | 47.11 | 47.19 | 47.05 | 47.14 | $47.12 \pm 0.05$ |
| Qwen3-8B | Decode | 45.68 | 45.75 | 45.68 | 45.69 | $45.70 \pm 0.03$ |

*Table 9.* LongBench average score of Sketch&Walk across four random seeds at 80% sparsity. Each per-seed entry is the mean over all 16 LongBench datasets.

The standard deviation across seeds stays at or below 0.05 points in every setting, well within the noise level of the underlying benchmarks, indicating that the randomized sketching in Sketch&Walk does not introduce meaningful run-to-run variance and that the accuracy reported in the main paper is robust to the choice of seed.

### D.3. Sensitivity to Block Size

The block size $B$ controls the granularity at which Sketch&Walk both estimates the block-level attention matrix and selects KV blocks for sparse attention. Smaller $B$ yields finer-grained selection but increases the number of blocks the walk must operate on, while larger $B$ amortizes more tokens per selected block at the cost of coarser relevance estimates. To examine this trade-off, we sweep $B \in \{16, 32, 64, 128\}$ on the multi-hop MUSIQUE dataset from LongBench, using Llama-3.1-8B-Instruct and Llama-3.2-1B-Instruct at 80% sparsity. Table 10 reports the resulting accuracy together with the dense-attention baseline for reference.

| Block size $B$ | Llama-3.1-8B-Instruct | Llama-3.2-1B-Instruct |
|---|---|---|
| Dense | 32.70 | 13.75 |
| 16 | **35.20** | 14.80 |
| 32 | 32.19 | 14.50 |
| 64 | 32.63 | **16.22** |
| 128 | 31.62 | 15.49 |

*Table 10.* Accuracy of Sketch&Walk on MUSIQUE as the block size $B$ varies, at 80% sparsity. Dense attention is included as a reference. Sketch&Walk matches or exceeds the dense baseline across all block sizes for both backbones.

Sketch&Walk matches or exceeds dense attention across all four block sizes for both backbones, with the best configuration even improving over dense by 2.5 points on Llama-3.1-8B-Instruct ($B = 16$) and by 2.5 points on Llama-3.2-1B-Instruct ($B = 64$). The accuracy spread across block sizes is at most 3.6 points on the 8B model and 1.7 points on the 1B model, indicating that Sketch&Walk is robust to this hyperparameter and that the default $B = 64$ used in our main experiments is a

---

**Algorithm 1** Sketch&Walk for prefill phase of layer $k$

---

**Input:** Query $\mathbf{Q}^k \in \mathbb{R}^{h \times n \times d}$, Key $\mathbf{K}^k \in \mathbb{R}^{h \times n \times d}$, Value $\mathbf{V}^k \in \mathbb{R}^{h \times n \times d}$ at layer $k$,
      block size $B$, reduced dimension $r$, top-$\tau$ blocks to select, sparsity exponent $s$, number of heads $h$,
      Hadamard matrix $\mathbf{H}_d \in \mathbb{R}^{d \times r}$, random walk state $R^{k-1} \in \mathbb{R}^{b_q \times b_k}$
**Output:** Attention output $\mathbf{O}^k \in \mathbb{R}^{h \times n \times d}$, updated random walk state $R^k$

{Prefill Stage: Block-wise Selection via Sketch-and-Walk + Sparse Attention on Original QKV}
{Step 1: Head-averaged Q/K for block selection}
$\mathbf{Q}_{\text{avg}}^k \leftarrow \frac{1}{h} \sum_{u=1}^{h} \mathbf{Q}^k[u, :, :] \in \mathbb{R}^{n \times d}$
$\mathbf{K}_{\text{avg}}^k \leftarrow \frac{1}{h} \sum_{u=1}^{h} \mathbf{K}^k[u, :, :] \in \mathbb{R}^{n \times d}$

{Step 2: Partition head-averaged Q/K into blocks (for sketching only)}
$b_q \leftarrow \lceil n/B \rceil$ {Number of query blocks}
$b_k \leftarrow \lceil n/B \rceil$ {Number of key blocks}
**for** $i = 1$ **to** $b_q$ **do**
    $\mathbf{Q}_{\text{avg}}^{k,(i)} \leftarrow \mathbf{Q}_{\text{avg}}^k[(i-1)B : iB, :] \in \mathbb{R}^{B \times d}$
**end for**
**for** $j = 1$ **to** $b_k$ **do**
    $\mathbf{K}_{\text{avg}}^{k,(j)} \leftarrow \mathbf{K}_{\text{avg}}^k[(j-1)B : jB, :] \in \mathbb{R}^{B \times d}$
**end for**

{Step 3: Compute block representatives (token-wise averaging within each block)}
**for** $i = 1$ **to** $b_q$ **do**
    $\bar{\mathbf{q}}_i^k \leftarrow \frac{1}{B} \sum_{t=1}^{B} \mathbf{Q}_{\text{avg}}^{k,(i)}[t, :] \in \mathbb{R}^d$
**end for**
**for** $j = 1$ **to** $b_k$ **do**
    $\bar{\mathbf{k}}_j^k \leftarrow \frac{1}{B} \sum_{t=1}^{B} \mathbf{K}_{\text{avg}}^{k,(j)}[t, :] \in \mathbb{R}^d$
**end for**
$\bar{\mathbf{Q}}^k \leftarrow [\bar{\mathbf{q}}_1^k; \ldots; \bar{\mathbf{q}}_{b_q}^k] \in \mathbb{R}^{b_q \times d}$
$\bar{\mathbf{K}}^k \leftarrow [\bar{\mathbf{k}}_1^k; \ldots; \bar{\mathbf{k}}_{b_k}^k] \in \mathbb{R}^{b_k \times d}$

{Step 4: Dimensionality reduction via Hadamard transform (on block representatives)}
$\tilde{\mathbf{Q}}^k \leftarrow \bar{\mathbf{Q}}^k \mathbf{H}_d \in \mathbb{R}^{b_q \times r}$
$\tilde{\mathbf{K}}^k \leftarrow \bar{\mathbf{K}}^k \mathbf{H}_d \in \mathbb{R}^{b_k \times r}$

{Step 5: Sketched block-level attention scores}
$\hat{\mathbf{A}}_{\text{block}}^k \leftarrow \tilde{\mathbf{Q}}^k (\tilde{\mathbf{K}}^k)^\top / \sqrt{r} \in \mathbb{R}^{b_q \times b_k}$

{Step 6: Sketch&Walk state update across layers}
$\mathbf{W}^k \leftarrow (\hat{\mathbf{A}}_{\text{block}}^k)^s$ {element-wise exponentiation}
**if** $k = 0$ **then**
    $R^0 \leftarrow \mathbf{W}^0$
**else**
    $R^k \leftarrow R^{k-1} \mathbf{W}^k$ {matrix multiplication}
**end if**

{Step 7: Select top-$\tau$ key blocks per query block}
**for** $i = 1$ **to** $b_q$ **do**
    $\mathcal{S}_i^k \leftarrow \text{TOPK-INDICES}(R^k[i, :], \tau)$
**end for**

{Step 8: Sparse attention on ORIGINAL per-head QKV using selected blocks}

**return** $\mathbf{O}^k, R^k$

---

---

**Algorithm 2** Sketch&Walk for decode phase of layer $k$

---

**Input:** New query token $\mathbf{Q}_t^k \in \mathbb{R}^{h \times d}$ at layer $k$, KV-cache $\mathbf{K}_{\text{cache}}^k \in \mathbb{R}^{h \times t \times d}$, $\mathbf{V}_{\text{cache}}^k \in \mathbb{R}^{h \times t \times d}$,
 cached query-block reps $\{\bar{\mathbf{q}}_i^k\}_{i=1}^{b_q}$, cached key-block reps $\{\bar{\mathbf{k}}_j^k\}_{j=1}^{b_k}$ with $\bar{\mathbf{q}}_i^k, \bar{\mathbf{k}}_j^k \in \mathbb{R}^d$,
 cached block-attn estimate $\hat{\mathbf{A}}_{\text{block,cache}}^k \in \mathbb{R}^{b_q \times b_k}$ from prefill,
 block size $B$, reduced dim $r$, top-$\tau$ blocks, sparsity exponent $s$, Hadamard matrix $\mathbf{H}_d \in \mathbb{R}^{d \times r}$,
 random walk state $R^{k-1} \in \mathbb{R}^{b_q \times b_k}$ from previous layer
**Output:** Attention output $\mathbf{O}_t^k \in \mathbb{R}^{h \times d}$, updated caches and random walk state $R^k$

{Step 1: Update KV-cache with new per-head key/value}
$\mathbf{K}_t^k, \mathbf{V}_t^k \leftarrow \text{PROJECT}(\mathbf{Q}_t^k) \ \{\mathbf{K}_t^k, \mathbf{V}_t^k \in \mathbb{R}^{h \times d}\}$
$\mathbf{K}_{\text{cache}}^k \leftarrow \text{APPEND}(\mathbf{K}_{\text{cache}}^k, \mathbf{K}_t^k)$
$\mathbf{V}_{\text{cache}}^k \leftarrow \text{APPEND}(\mathbf{V}_{\text{cache}}^k, \mathbf{V}_t^k)$

{Step 2: Update head-averaged key block representative for sketching}
$b_{\text{curr}} \leftarrow \lceil \frac{t}{B} \rceil$ {current key-block index}
$\mathbf{k}_{t,\text{avg}}^k \leftarrow \frac{1}{h} \sum_{u=1}^h \mathbf{K}_t^k[u,:] \in \mathbb{R}^d$
**if** $t \bmod B = 1$ **then**
 $\bar{\mathbf{k}}_{b_{\text{curr}}}^k \leftarrow \mathbf{k}_{t,\text{avg}}^k$
 $c_{b_{\text{curr}}} \leftarrow 1$
**else**
 $\bar{\mathbf{k}}_{b_{\text{curr}}}^k \leftarrow \frac{c_{b_{\text{curr}}} \bar{\mathbf{k}}_{b_{\text{curr}}}^k + \mathbf{k}_{t,\text{avg}}^k}{c_{b_{\text{curr}}} + 1}$
 $c_{b_{\text{curr}}} \leftarrow c_{b_{\text{curr}}} + 1$
**end if**

{Step 3: Head-averaged query for sketching + Hadamard reduction}
$\mathbf{q}_{t,\text{avg}}^k \leftarrow \frac{1}{h} \sum_{u=1}^h \mathbf{Q}_t^k[u,:] \in \mathbb{R}^d$
$\tilde{\mathbf{q}}_t^k \leftarrow \mathbf{q}_{t,\text{avg}}^k \mathbf{H}_d \in \mathbb{R}^r$
$\tilde{\bar{\mathbf{K}}}^k \leftarrow [\bar{\mathbf{k}}_1^k \mathbf{H}_d; \dots; \bar{\mathbf{k}}_{b_{\text{curr}}}^k \mathbf{H}_d] \in \mathbb{R}^{b_{\text{curr}} \times r}$
$\hat{\mathbf{a}}_{\text{new}}^k \leftarrow \tilde{\mathbf{q}}_t^k (\tilde{\bar{\mathbf{K}}}^k)^\top / \sqrt{r} \in \mathbb{R}^{1 \times b_{\text{curr}}}$

{Step 4: Update cached block-attn estimate with the new row *and* last column}
$b_q \leftarrow \lceil \frac{t}{B} \rceil$ {current query-block index}
$\hat{\mathbf{A}}_{\text{block}}^k \leftarrow \hat{\mathbf{A}}_{\text{block,cache}}^k$
{(i) Update the new query row}
$\hat{\mathbf{A}}_{\text{block}}^k[b_q, 1{:}b_{\text{curr}}] \leftarrow \hat{\mathbf{a}}_{\text{new}}^k$
{(ii) Update the last column for the current key block}
$\tilde{\bar{\mathbf{Q}}}^k \leftarrow [\bar{\mathbf{q}}_1^k \mathbf{H}_d; \dots; \bar{\mathbf{q}}_{b_q}^k \mathbf{H}_d] \in \mathbb{R}^{b_q \times r}$
$\tilde{\bar{\mathbf{k}}}_{b_{\text{curr}}}^k \leftarrow \bar{\mathbf{k}}_{b_{\text{curr}}}^k \mathbf{H}_d \in \mathbb{R}^r$
$\hat{\mathbf{c}}_{\text{new}}^k \leftarrow \tilde{\bar{\mathbf{Q}}}^k \tilde{\bar{\mathbf{k}}}_{b_{\text{curr}}}^k / \sqrt{r} \in \mathbb{R}^{b_q \times 1}$
$\hat{\mathbf{A}}_{\text{block}}^k[1{:}b_q, b_{\text{curr}}] \leftarrow \hat{\mathbf{c}}_{\text{new}}^k$

{Step 5: Random Walk}
$\mathbf{W}^k \leftarrow \left(\hat{\mathbf{A}}_{\text{block}}^k\right)^s$
**if** $k = 0$ **then**
 $R^0 \leftarrow \mathbf{W}^0$
**else**
 $R^k \leftarrow R^{k-1} \mathbf{W}^k$
**end if**

{Step 6: Select top-$\tau$ blocks using the new query row of the walk (always include current block)}
$\mathcal{S} \leftarrow \text{TOPK-INDICES}(R^k[b_q, 1{:}b_{\text{curr}}], \tau - 1) \cup \{b_{\text{curr}}\}$

{Step 7: Sparse attention on ORIGINAL per-head QKV over selected blocks}
Initialize $\mathbf{O}_t^k \leftarrow \mathbf{0}^{h \times d}$
**for** $u = 1$ **to** $h$ **do**
 $\mathbf{K}_{\text{sel}}^{k,(u)} \leftarrow \text{GATHERBLOCKS}(\mathbf{K}_{\text{cache}}^k[u,:,:], \mathcal{S}, B)$
 $\mathbf{V}_{\text{sel}}^{k,(u)} \leftarrow \text{GATHERBLOCKS}(\mathbf{V}_{\text{cache}}^k[u,:,:], \mathcal{S}, B)$
 $\mathbf{a}^{k,(u)} \leftarrow \text{softmax}\left(\mathbf{Q}_t^k[u,:](\mathbf{K}_{\text{sel}}^{k,(u)})^\top / \sqrt{d}\right)$
 $\mathbf{O}_t^k[u,:] \leftarrow \mathbf{a}^{k,(u)} \mathbf{V}_{\text{sel}}^{k,(u)}$
**end for**

$\hat{\mathbf{A}}_{\text{block,cache}}^k \leftarrow \hat{\mathbf{A}}_{\text{block}}^k$ {persist updated block estimate}
**return** $\mathbf{O}_t^k$, updated $\mathbf{K}_{\text{cache}}^k, \mathbf{V}_{\text{cache}}^k, \{\bar{\mathbf{q}}_i^k\}, \{\bar{\mathbf{k}}_j^k\}, \hat{\mathbf{A}}_{\text{block,cache}}^k, R^k$

---

reasonable trade-off between selection granularity and walk efficiency.

### D.4. Contribution of the Walk on Multi-Hop Tasks

To directly verify that the walk component is responsible for resolving multi-hop dependencies, we run a controlled study on the Variable Tracking (VT) task from RULER (Hsieh et al., 2024), in which the model must follow chains of variable assignments (e.g., $A = v \to B = A \to C = B \to \cdots$) under long-context inputs. We compare Sketch&Walk against a *Sketch-only* variant that selects blocks from the raw sketch scores $\tilde{A}^k_{\text{block}}$ without the walk propagation, with both methods evaluated on Qwen3-8B (Yang et al., 2025) at 20% sparsity. As shown in Table 11, the walk mechanism consistently outperforms Sketch-only, and the gap *widens* at longer sequence lengths where multi-hop chains are more spread out, confirming Sketch&Walk's advantage in capturing multi-hop dependencies.

| Seq. length | 4K | 8K | 16K | 32K | 64K |
|---|---|---|---|---|---|
| Sketch&Walk | **100.0** | **100.0** | **100.0** | **98.8** | **98.8** |
| Sketch-only | 95.8 | 93.6 | 96.6 | 86.6 | 86.6 |

*Table 11.* Variable-Tracking (VT) accuracy on Qwen3-8B at 20% sparsity. The walk component yields larger gains at longer contexts, where multi-hop chains are harder to resolve from single-layer scores.

## E. Limitations

**Walk-state memory overhead.** A genuine overhead of Sketch&Walk is the memory required to store the sketch-determined walk state $R^k$, which is a $b \times b$ block-level matrix where $b = \lceil n/B \rceil$ is the number of blocks. Concretely, the walk state scales as $\mathcal{O}\big((n/B)^2\big)$, so the overhead is controllable through the block size $B$: doubling $B$ reduces the walk-state size by $4\times$. As a reference point, at a context length of 128K with $B = 64$ the walk state is approximately 500 MB; increasing the block size to $B = 256$ brings this down to roughly 32 MB. Meanwhile, the KV cache at 128K for Llama-3.1-8B-Instruct is approximately 16 GB, so with an appropriately scaled block size the walk-state overhead represents well under 1% of total inference memory. Our ablation in Appendix D.3 further shows that Sketch&Walk's accuracy remains close to dense attention across $B \in \{16, 32, 64, 128\}$, indicating that scaling $B$ at very long contexts is a practical knob rather than a fundamental constraint.

**Randomized sketching.** Sketch&Walk relies on randomized Hadamard sketching, so its theoretical guarantees are probabilistic rather than worst-case deterministic. The seed-stability study in Appendix D.2 shows that the resulting accuracy variation is at or below 0.05 points on LongBench across four seeds, well within benchmark noise, but the formal bounds in Appendix B.3.1 still hold only with high probability over the sketch.

**Hyperparameter selection.** Sketch&Walk introduces three hyperparameters: the sketch dimension $k$, the walk degree $s$, and the block size $B$. Our ablations show that accuracy is robust over a broad range of each (Section 4, Appendices D.3 and D.4), and the defaults used in the main paper transfer across the four backbones we tested. Nevertheless, the optimal triple may shift modestly on substantially different model families or task distributions, and we have not performed an exhaustive joint sweep.

**Scope of the efficiency measurements.** Our efficiency analysis (Section 4) reports single-GPU, single-batch kernel-level wall-clock speedups against FlashAttention-2 on H100/H200 hardware. Multi-GPU, batched-serving, or quantized-inference deployments may shift the relative ordering of methods, and end-to-end model-level speedups will generally be smaller than the kernel-level numbers because attention is only a fraction of total inference cost at very long context.

