# OpenReview forum: "Scout Before You Attend: Sketch-and-Walk Sparse Attention for Efficient LLM Inference"
_ICML.cc/2026/Conference — ICML 2026 regular_

### Official Review · Reviewer_iigv · 2026-03-04

**Soundness:** 3
**Presentation:** 3
**Significance:** 3
**Originality:** 3
**Overall Recommendation:** 5
**Confidence:** 4

**Summary:**

This paper proposes Sketch&Walk, a training-free sparse attention algorithm for efficient LLM inference across both prefill and decode phases. It addresses a key flaw in existing block-sparse methods: relying on single-layer attention scores often prematurely discards tokens crucial for multi-hop reasoning in deeper layers. To solve this, the authors introduce a two-stage mechanism: first, Small-World Sketching uses Hadamard projections to cheaply estimate block-level attention scores ; second, a Sketch-Determined Walk aggregates these estimates across layers via matrix multiplication to capture multi-hop dependencies and guide block selection. Evaluations on LongBench and RULER demonstrate near-lossless accuracy at 80% sparsity, achieving up to 6x prefill and 1.6x decode speedups.

**Compliance With Llm Reviewing Policy:**

Affirmed.

**Final Justification:**

I thank the authors for the detailed response. All my concerns are addressed. I'll raise my score from 4 to 5.

**Key Questions For Authors:**

1. Does Sketch&Walk allow for the actual eviction/dropping of unselected KV cache blocks from GPU memory during the decode phase, or does it merely skip the compute while keeping the full KV cache resident in memory? Since decode is heavily memory-bandwidth bound, clarifying this is crucial for real-world deployment.
2. While the theoretical analysis (Lemmas 2.4, 2.6) explicitly relies on the block size $B$ to bound sketching errors and guarantee selection accuracy, the experiments fix $B=64$ without any empirical ablation. Could the authors provide an ablation study on different block sizes to demonstrate the practical trade-off between finer-grained sparsity and sketching accuracy?

**Limitations:**

The authors should elaborate on the practical hardware/memory limitations of their approach.

**Strengths And Weaknesses:**

Pros:

- The identification of the "multi-hop dependency breakdown" caused by aggressive layer-wise sparsity is a highly insightful observation. Solving this via an exponentiated cross-layer Markov walk ($R^k$) applied entirely at inference time is an elegant and well-motivated conceptual leap.
- The paper provides comprehensive theoretical guarantees. The authors mathematically bound the sketching approximation error (Lemmas 2.4, 2.6) and formally demonstrate how the random walk maintains the global information structure under heavy-tailed attention assumptions (Lemma 2.7).
- Sketch&Walk delivers strong empirical performance. Achieving near-lossless results at 80% sparsity across challenging datasets like RULER and LongBench on modern architectures (Llama-3.1, Qwen-2) proves its practical viability.

Cons:

- The paper heavily emphasizes computational speedup (latency/throughput) but glosses over the exact memory implications during the decode phase. Because the block selection state ($R^k$) relies on cross-layer interactions, it implies the system might still need to retain the *full* uncompressed KV cache in VRAM to perform the exact sparse attention once the blocks are selected. If so, this mitigates compute bottlenecks but not memory-bandwidth bottlenecks (which typically dominate decode).
- The baselines and models are outdated. More recent sparse attention methods are not included as baselines or at least discussed, for example, SpargeAttn[1], XAttention[2], PBS-Attn[3] for prefill. The paper also uses models released 2 years ago(Llama 3.1/Qwen2) instead of modern models like Qwen3.

[1] https://arxiv.org/abs/2502.18137

[2] https://arxiv.org/abs/2503.16428

[3] https://arxiv.org/abs/2510.21270

---

> ### Author Rebuttal · Authors · 2026-03-31
>
> ### **`W1, Q1 - does Sketch&Walk evict KV cache blocks or only skip compute?`**
>
> We thank the reviewer for this question and clarify two points.
>
> **On memory storage:** We acknowledge that Sketch&Walk retains the full KV cache in VRAM and does not reduce memory footprint, **like other KV cache selection and prefilling acceleration methods**. Sketch&Walk provides acceleration with no memory saving, with the benefit of minimal performance loss.
>
> **On memory bandwidth and compute:** We respectfully clarify the reviewer's statement that Sketc&hWalk:
>
> "*mitigates compute bottlenecks but not memory-bandwidth bottlenecks (which typically dominate decode)*."
>
> Memory bandwidth refers to the *transfer time* of loading KV blocks from VRAM into compute units, **not the storage footprint**. Sketch&Walk directly addresses this:
>
> - At each decode step, only the selected sparse blocks are transferred for attention computation, rather than the full KV cache, reducing the volume of data transferred proportionally to the sparsity ratio. Since attention kernels must load KV blocks from HBM before computation, reducing the number of accessed blocks directly reduces memory traffic. Therefore, the observed latency improvements in decode (which is known to be memory-bandwidth bound) imply proportional reductions in KV cache memory-bandwidth usage.
> - Prefilling is *compute*-bounded rather than memory-bounded, a well-established characterization of the prefill regime. Sketch&Walk addresses this by restricting attention computation to a sparse block set, reducing FLOPs (and memory transfer) accordingly.
>
>
> **On Sketch&Walk memory overhead:** We acknowledge that Sketch&Walk introduces additional memory due to the walk matrix. However, this overhead is minimal in practice. We verify this empirically below: at 16K context length, Sketch&Walk adds only 1.26 GB over the dense baseline, confirming the additional cost is negligible.
>
> | Method | Peak Memory (GB) |
> |---|---|
> | Dense | 19.17 |
> | Sketch&Walk | 20.43 |
>
> In summary, Sketch&Walk reduces *memory-bound* during decode and *compute-bound* during prefill, while introducing minimal memory overhead beyond the standard KV cache and incurring minimal performance degradation.
>
>
> ### **`W2 - Baselines outdated, models are old`**
>
> We now compare SpargeAttn and XAttention at 80% (more extreme) sparsity to align with Sketch&Walk's setting. We report the average across 16 LongBench datasets, and Sketch&Walk is better than both baselines on both models:
>
> | Method | Llama-3.1-8B | Llama-3.2-1B |
> |---|---|---|
> | Sketch&Walk | 47.67 | 31.01 |
> | SpargeAttn | 46.34 | 28.70 |
> | XAttention | 35.45 | 25.07 |
>
> We further conduct experiments with **Qwen3-8B** on LongBench with Sktetch&Walk and the suggested baselines at 80% sparsity. Sketch&Walk closely matches the dense baseline while being better than other prefilling methods:
>
> | Method | LB Avg |
> |---|---|
> | Dense | 48.02 |
> | Sketch&Walk| 47.12 |
> | XAttention | 32.51 |
> | SpargeAttn | 46.67 |
>
>
>
>
> ### **`Q2 - Ablation on block size B`**
>
> We ablate block sizes B ∈ {16, 32, 64, 128} on the musique dataset:
>
> | B | Llama-3.1-8B | Llama-3.2-1B |
> |---|---|---|
> | Dense | 32.70 | 13.75 |
> | 16 | 35.20 | 14.80 |
> | 32 | 32.19 | 14.50 |
> | 64 | 32.63 | 16.22 |
> | 128 | 31.62 | 15.49 |
>
> Sketch&Walk is robust across all block sizes, consistently matching or exceeding the dense baseline. Interestingly, smaller block sizes sometimes push performance above dense.
>
> *Note: our new experiments run on H200; numbers might differ slightly from the paper's H100 results.*
>
>
> We hope these additional clarifications and experiments address the reviewer’s concerns. If the reviewer finds them helpful, we would greatly appreciate consideration for a higher rating.

---

> > ### Author Rebuttal · Reviewer_iigv · 2026-04-02
> >
> > I appreciate the authors' effort in providing additional experiments. The memory clarification and block size ablation are helpful. However, a few concerns remain:
> > 1. Decode acceleration is overclaimed. The paper repeatedly emphasizes uniform applicability to both prefill and decode, but the actual decode speedup is marginal (near 1.0× at 16K, only 1.6× at 128K). No latency comparison against decode baselines (Quest/Adamas) is provided — only accuracy. The per-step walk overhead (b×b matmul per layer) is non-trivial for latency-sensitive decoding.
> > 2. Memory overhead understated. The reported 1.26 GB overhead is at 16K (b=256). At 128K (b=2048), the walk state alone scales to ~500 MB across layers, which is non-negligible relative to the KV cache.
> > 3. Not all suggested baselines were well addressed in the rebuttal. The XAttention scores are far from the original paper and the more recent PBS-Attn is not mentioned.
> >
> > I keep my score at 4 for now and welcome further discussion on the remaining points.

---

> > > ### Author Response · Authors · 2026-04-03
> > >
> > > ### **`Q1 - Decode acceleration and walk overhead`**
> > > We appreciate the reviewer raising this point. We would like to clarify that Figure 4 is a decode-only setting, where we run *full dense attention during prefill plus the additional overhead of computing the walk matrix*, in order to test the efficiency of Sketch&Walk in the decode phase only, not prefill + decode scenario.
> > >
> > > In this particular setting, Sketch&Walk *cannot benefit from sparse prefill*, yet the walk matrix had to be computed as *additional overhead* on top of a full prefill, without any sparse prefill savings to offset it.
> > >
> > > One might fix this by incorporating prefill acceleration into the comparison — but that would disadvantage Quest and Adamas instead, since they provide no prefill acceleration. Any single-phase comparison therefore favors one side by design:
> > >
> > > - Decode-only (Figure 4): walk overhead charged to decode with no prefill savings → disadvantages Sketch&Walk
> > > - Full pipeline: Sketch&Walk accelerates both prefill and decode, so Sketch&Walk will **definitively outperform decode-only methods**, making the comparison unfair to Quest and Adamas
> > >
> > > That is why we did not compare with Quest or Adamas directly. We now present a comparison of decode speedup (excluding overhead from prefill) and prefill+decode speedup across all three methods on Llama-3.1-8B (H200, FP16), using kernels from their publicly released repositories:
> > >
> > > | Context | SW P+D | SW Dec | Quest P+D | Quest Dec | Adamas P+D | Adamas Dec |
> > > |--------:|:------:|:------:|:---------:|:---------:|:----------:|:----------:|
> > > | 8K      | 1.42×  | 1.36×  |   0.89×   |   1.40×   |   0.68×    |   0.95×    |
> > > | 16K     | 2.38×  | 1.82×  |   1.62×   |   2.05×   |   0.92×    |   0.91×    |
> > > | 32K     | 3.53×  | 2.54×  |   1.47×   |   2.48×   |   0.83×    |   0.79×    |
> > > | 64K     | 3.93×  | 3.27×  |   1.25×   |   2.57×   |   0.86×    |   0.77×    |
> > > | 96K     | 4.28×  | 3.69×  |   1.17×   |   2.72×   |   0.88×    |   0.73×    |
> > >
> > > Sketch&Walk's decode and prefill+decode speedups scale consistently with context. Quest achieves strong decode speedup, but its prefill+decode gain is limited since it does not accelerate prefill. Adamas does not reach positive speedup in our setting; we believe their paper evaluates on A6000 GPUs, whereas H200's higher memory bandwidth makes the dense baseline harder to beat.
> > >
> > > **Regarding the per-step walk overhead:**
> > > - During decode, the walk reduces to a row-vector × b×b matrix multiply per layer, where $b = n/B$ and $B$ is the block size. The walk costs $O(n^2/B^2)$ FLOPs, while dense decode attention costs $O(H \cdot n \cdot d)$. Their ratio $n/(B^2 \cdot H \cdot d)$ is under 1% at 96K context with the paper's default settings.
> > > - We refer the reviewer to Figure 5, which empirically confirms the walk overhead is negligible compared to attention across sequence lengths and budgets.
> > >
> > > ### **`Q2 - Memory Overhead`**
> > > We note that the 1.26 GB overhead is evaluated at block size 64 with a 16K context length. Under a 128K context, one can increase the block size to reduce the number of blocks. Since the walk matrix scales as the square of the number of blocks, increasing the block size offers a quadratic reduction in walk state memory. To verify that performance is preserved under larger block sizes, we evaluate Llama-3.1-8B with block sizes 128 and 256 on LongBench:
> > >
> > > | Block Size | Avg |
> > > |---|---|
> > > | 64 | 47.67 |
> > > | 128 | 46.73 |
> > > | 256 | 47.54 |
> > >
> > > Performance remains stable across block sizes, suggesting that Sketch&Walk can trade block granularity for memory efficiency at longer contexts with negligible impact on quality.
> > >
> > > We *will acknowledge in limitation section* that walk state memory is a genuine overhead of Sketch&Walk. However, this overhead is controllable via block size: at longer contexts, we can adopt larger block sizes. Since walk state scales as $O(b^2) = O(n^2/B^2)$, doubling $B$ reduces walk state by $4\times$. Taking the reviewer's estimate of ~500 MB at 128K with $B=64$, increasing to $B=256$ brings this to ~32 MB. Meanwhile, the KV cache at 128K for Llama-3.1-8B is approximately 16 GB, so with appropriately scaled block size, the walk state represents well under 1% of total memory.
> > >
> > > ### **`Q3 - Baselines`**
> > > We thank the reviewer for pointing this out, and we apologize for the oversight. Upon careful investigation, we discovered an implementation error in our XAttention evaluation: XAttention's threshold $\tau$ controls cumulative attention mass coverage, not sparsity fraction. We mistakenly interpreted τ as a sparsity fraction and set threshold=0.2, which retains only 20% of attention mass.
> > >
> > > We have now re-evaluated XAttention using the paper's original per-layer thresholds, and additionally include PBS-Attn. The corrected results are:
> > >
> > > | Model | Sketch&Walk | XAttention | PBS-Attn |
> > > |---|---|---|---|
> > > | Llama-3.1-8B | 47.67 | 47.05 | 46.80 |
> > > | Llama-3.2-1B | 31.01 | 30.29 | 30.02 |
> > >
> > > We will update the paper to reflect all of your suggested baselines

---

### Official Review · Reviewer_K6rt · 2026-03-04

**Soundness:** 3
**Presentation:** 3
**Significance:** 2
**Originality:** 3
**Overall Recommendation:** 4
**Confidence:** 4

**Summary:**

The Sketch&Walk framework introduces a training-free sparse attention mechanism designed to optimize large language model inference during both pre-filling and decoding phases. This method utilizes block-level sketching to approximate attention scores through Hadamard projections which reduces computational costs. A deterministic walk mechanism aggregates these estimates across multiple layers to identify significant attention relationships and dependencies between tokens. By selecting only the most important attention blocks based on these accumulated scores, the approach maintains high accuracy at low attention densities. Empirical evaluations demonstrate that this technique achieves significant speedup in throughput while preserving performance on long-context benchmarks.

**Compliance With Llm Reviewing Policy:**

Affirmed.

**Final Justification:**

The author's rebuttal has adequately addressed my concerns.

**Key Questions For Authors:**

In Figure 1, within the first section titled "1. Small-World Sketching (SWS)", there appears to be a potential typo in the text description of the orange rectangle located at the bottom right of the "token space sketching" arrow. The current notation indicates $\bar{\mathbf{Q}} \in \mathbb{R}^{b \times d}$, but should it correctly be $\bar{\mathbf{K}} \in \mathbb{R}^{b \times d}$ to reflect the key blocks being sketched?

**Limitations:**

yes

**Strengths And Weaknesses:**

Strengths:
- The Sketch&Walk framework provides a unified solution that accelerates both the pre-filling and decoding phases of large language model inference, demonstrating versatility across different stages of the generation process.
- The manuscript is well-structured and presents extensive experimental evaluations across various models and benchmarks. Additionally, the authors provide a solid theoretical analysis to justify the effectiveness and convergence of the proposed mechanism.

----

Weaknesses:
- The Sketch&Walk framework lacks conceptual novelty because its primary components are already established in efficient LLM inference research. The Small-World Sketching module utilizes block-level importance estimation through low-dimensional projections, which is a standard method in sparse attention literature. Similarly, the Walk mechanism aggregates scores across layers to capture multi-hop dependencies, but this approach closely resembles existing cross-layer propagation techniques like TidalDecode [1]. The authors should provide a rigorous discussion of their fundamental motivation to distinguish this work from prior art.

----

[1] Yang L, Zhang Z, Chen Z, et al. Tidaldecode: Fast and accurate llm decoding with position persistent sparse attention[J]. arXiv preprint arXiv:2410.05076, 2024.

---

> ### Author Rebuttal · Authors · 2026-03-31
>
> We thank the reviewer for the initial positive rating of the work. We believe some of the weaknesses raised may be due to confusion, and we hope to address your concerns below.
>
> ### **`W1 - Conceptual novelty`**
>
>
> We respectfully disagree with this characterization of our work. We believe this assessment might differ from the evaluations of the other three reviewers who independently assessed the novelty of Sketch&Walk:
> - R `aKiX` wrote: *"This is an interesting and novel approach. It is nice to look at the multi-hop interactions between tokens across different layers, as opposed to simply looking at single-hop interactions."*
> - R `iigv` described the Walk mechanism as *"an elegant and well-motivated conceptual leap"* and called the identification of the multi-hop dependency breakdown *"a highly insightful observation."*
> - R `yHcU` noted that *"the theoretical contributions are non-trivial"* and that *"the paper addresses this exact problem elegantly."*
>
> ----
>
> We address each point below and hope to clarify the distinctions from prior art.
>
> **Key distinction (multi-hop token connection):**
> - Prior sparse attention methods select blocks using **per-layer (one-hop) attention scores** (e.g., top-$k$ based on $QK^\top$), or reuse tokens that consistently receive high attention across layers.
> - As discussed in Sec. 1 and Sec. 3, this design fails when dependencies only emerge through *composition of attention across layers* (*multi-hop dependency*).
> - In such cases, a block may *not be selected by any earlier layer*, yet becomes important after multi-layer composition, and thus cannot be recovered by layer-wise selection.
>
> **What Sketch&Walk adds (cross-layer composition).**
> - Sketch&Walk maintains a walk state $R_k$ that accumulates block-level attention estimates across layers. Block selection is then based on $R_k$, rather than per-layer scores.
> - This enables recovery of **blocks whose importance emerges only through attention composition across layers**, which are missed by existing methods.
> - We empirically validate this behavior in our empirical study on multi-hop experiments (please refer to our response to W2 of R`yHcU` and W1 of R`aKiX`).
>
> ------
>
> **On the Sketch:** While low-dimensional projections are used in prior work, Sketch&Walk employs Hadamard-based sketching with order preservation, with provable guarantees on error bound.
>
> **Walk vs. TidalDecode.** We clarify the distinction with TidalDecode as follows:
> - TidalDecode identifies position-persistent attention patterns and reuses static token selections across layers to reduce redundant KV loading: **a layer-wise cache selection strategy**.
> - The Walk in Sketch&Walk instead performs a **walk on the block-level attention sketch**, propagating importance scores to capture transitive, multi-hop dependencies that single-layer scores would miss.
> - Therefore, TidalDecode focuses on **reusing consistently selected tokens**, while Sketch&Walk targets **blocks that are not selected by earlier layers (if by default) but become important through multi-layer composition**.
> - This conceptual difference is reflected empirically: under the same setting, Sketch&Walk consistently outperforms TidalDecode across both model scales:
>
> | Model | TidalDecode | Sketch&Walk |
> |-------|------------:|------------:|
> | Llama3.2-1B | 28.73 | 30.34 |
> | Llama3.1-8B | 47.19 | 48.00 |
>
> We hope this clarifies that Sketch&Walk addresses a distinct limitation of existing sparse attention methods, rather than being a direct combination of prior components.
>
>
> ### **`Q1 - Typo in Figure 1`**
> We thank the reviewer for your careful read. We would update Figure  in the updated version.
>
>
> We hope these clarifications better highlight the merit and novelty of our work, and may merit consideration for a higher rating.

---

> > ### Author Rebuttal · Reviewer_K6rt · 2026-04-01
> >
> > I appreciate the authors' efforts in addressing my comments. All my concerns have been resolved.

---

> > > ### Author Response · Authors · 2026-04-07
> > >
> > > Dear Reviewer K6rt,
> > >
> > > Thank you for your time and careful reading. We appreciate your recognition of the unified framework and the solid theoretical analysis. We are glad that our clarification on the key distinction between Sketch&Walk and TidalDecode has resolved the novelty concern. We are glad all concerns have been addressed and thank you for your continued positive rating.
> > >
> > > Warm regards,
> > >
> > > *Paper21167 Authors*

---

### Official Review · Reviewer_yHcU · 2026-03-12

**Soundness:** 3
**Presentation:** 3
**Significance:** 3
**Originality:** 3
**Overall Recommendation:** 4
**Confidence:** 3

**Summary:**

This paper introduces Sketch&Walk Attention, an efficient, multi-hop interaction-aware, and training-free sparse attention mechanism designed to accelerate Large Language Model (LLM) inference. By leveraging randomized sketching and deterministic walks, the method aggregates attention scores across layers to capture high-order relationships without explicitly computing the full dense attention matrix. The authors provide a theoretical analysis of the error bounds and computational efficiency under mild assumptions. Empirically, the paper evaluates Sketch&Walk against vanilla dense attention and other sparse attention baselines, demonstrating its strong potential as a highly efficient, plug-and-play substitute.

**Compliance With Llm Reviewing Policy:**

Affirmed.

**Key Questions For Authors:**

1. Do the authors have any preliminary empirical results on synthetic multi-hop reasoning tasks that can directly visualize or quantify the model's superiority in capturing high-order relations over standard single-hop sparse attention?
2. Could the authors provide the standard deviations for their main empirical results across multiple random seeds?

**Limitations:**

Yes

**Strengths And Weaknesses:**

**Strengths**
1. **Excellent Presentation Quality:** The paper is exceptionally well-written and structured. The mathematical notation is clean, the proofs are clear and easy to verify, and the visual aids effectively illustrate the sketch-and-walk aggregation mechanism, making a complex algorithmic process highly accessible.
2. **Strong Theoretical Foundations:** The theoretical contributions are non-trivial. The authors establish solid error bound guarantees and efficiency metrics under mild assumptions, providing strong mathematical justification for the sketching approximations.
3. **Highly Important Problem and Motivation:** Capturing high-order (multi-hop) relations efficiently is a critical bottleneck in deploying long-context LLMs. The paper addresses this exact problem elegantly, and the empirical results convincingly support the claims of accelerated inference with minimal performance degradation.

**Weaknesses**
1. **Formatting of Theoretical Claims (Minor):** Corollary B.10 is stated as a corollary, but it does not read as an explicit mathematical statement. Instead, it serves as an implication or qualitative discussion of the preceding theoretical results. This should ideally be reformatted as a "Remark" or integrated into the standard text discussion rather than being labeled as a Corollary.
2. **Lack of Synthetic Validation for High-Order Relations:** While the empirical results demonstrate strong end-to-end downstream performance, they do not perfectly isolate the core theoretical claim: capturing high-order/multi-hop relations. The paper would be significantly strengthened by including a controlled synthetic experiment (e.g., a multi-hop graph reasoning task) to explicitly demonstrate the mechanism's effectiveness at resolving multi-hop dependencies compared to baseline sparse attention.
3. **Statistical Variance and Significance:** Because the proposed method relies heavily on randomized sketching algorithms, the stability of the approximation is a potential concern. The experiments would be much more trustworthy if they were repeated across multiple random seeds, with the standard deviations explicitly reported to ensure the statistical significance of the reported performance.

---

> ### Author Rebuttal · Authors · 2026-03-31
>
> ### **`W1 - Corollary B.10 should be reformatted as a "Remark"`**
>
> We thank the reviewer for this suggestion. We would update B.10 to be a Remark in the updated version.
>
> ### **`W2, Q1 - Lack of empirical validation for multi-hop relations.`**
>
> We kindly note that our experiments feature multi-hop datasets such as MuSiQue and VT (RULER), though their scores might be condensed into an aggregate average in the paper. Here we present a controlled VT experiment with Qwen3-8B to directly verify Sketch&Walk's ability to capture multi-hop dependencies. VT is a synthetic benchmark where the model must follow chains of variable assignments (e.g., `A=v → B=A → C=B → ... → E=D`) under long-context setting (4k - 64k).
>
> We compare **Sketch&Walk** against **Sketch-only** (block selection based solely on raw sketch scores $\mathbf{p} = \text{Sketch}(Q, K)$, without the walk propagation $\mathbf{p}^{(t)} = \mathbf{p}^{(t-1)} \cdot \mathbf{A}_{\text{blk}}^s$), both at 20% sparsity on Qwen3-8B:
>
> | Seq. length | 4K | 8K | 16K | 32K | 64K |
> |---|---|---|---|---|---|
> | Sketch&Walk | 100.0 | 100.0 | 100.0 | 98.8 | 98.8 |
> | Sketch-only | 95.8 | 93.6 | 96.6 | 86.6 | 86.6 |
>
> The walk mechanism consistently outperforms sketch-only, with the gap widening at longer sequences where multi-hop chains are more spread out — directly demonstrating Sketch&Walk's advantage in resolving multi-hop dependencies. We also refer the reviewer to our response to W1 of R`aKiX`, where we analyze the attention pattern to empirically verify the multi-hop ability of Sketch&Walk from another angle.
>
>
> ### **`W3, Q2 - Statistical variance: report std across multiple random seeds`**
>
> We thank the reviewer for this suggestion. We have run Sketch&Walk across 4 random seeds on all 16 LongBench datasets for both Llama-3.2-1B-Instruct and Qwen3-8B, reporting the average score per seed and the mean$\pm$std below.
>
> | Model | Mode | seed0 | seed1 | seed2 | seed3 | Mean$\pm$Std |
> |---|---|---|---|---|---|---|
> | Llama-3.2 | Prefilling | 30.69 | 30.81 | 30.75 | 30.81 | 30.76 $\pm$ 0.05 |
> | Llama-3.2| Decoding | 30.42 | 30.29 | 30.31 | 30.32 | 30.34 $\pm$ 0.05 |
> | Qwen3 | Prefilling | 47.11 | 47.19 | 47.05 | 47.14 | 47.12 $\pm$ 0.05 |
> | Qwen3 | Decoding | 45.68 | 45.75 | 45.68 | 45.69 | 45.70 $\pm$ 0.03 |
>
> The standard deviation is $\leq$ 0.05 across all settings, confirming that Sketch&Walk is highly stable despite its randomized nature. We will incorporate this discussion into our appendix in the updated version to further demonstrate the robustness of Sketch&Walk.
>
> We hope the above responses have addressed all of the reviewer's concerns. We are glad the reviewer found our work elegant and strong. Please excuse us for asking too much — but may we kindly request the reviewer to consider a higher rating in light of the additional experiments and clarifications provided?

---

> > ### Author Rebuttal · Reviewer_yHcU · 2026-04-01
> >
> > I truly appreciate the author's effort in responding to my comments. I think my concerns are resolved, and I would like to keep my positive recommendation.

---

> > > ### Author Response · Authors · 2026-04-07
> > >
> > > Dear Reviewer yHcU,
> > >
> > > Thank you for your time and detailed review. We are grateful for your recognition of the theoretical contributions and the importance of the problem we address. Your suggestions on synthetic multi-hop validation and statistical variance were helpful, and we are glad our responses have fully resolved your concerns. We appreciate your continued positive recommendation.
> > >
> > > Warm regards,
> > >
> > > *Paper21167 Authors*

---

### Official Review · Reviewer_aKiX · 2026-03-12

**Soundness:** 3
**Presentation:** 3
**Significance:** 3
**Originality:** 3
**Overall Recommendation:** 4
**Confidence:** 5

**Summary:**

This paper proposes a training free sparse attention mechanism that uses Hadamard sketching to get an approximation of attention scores and uses an inter-layer walk mechanism to select top-k attention blocks. The method shows high quality as low as 20% density, and the authors report up to 6x inference speedup. The key insight is computing “multi-hop” token connections.

**Compliance With Llm Reviewing Policy:**

Affirmed.

**Final Justification:**

I am keeping my positive score after a strong rebuttal.

**Key Questions For Authors:**

* Can you do some analysis of the attention matrices to measure how often multi-hop connections are necessary? Some empirical analysis beyond the theoretical bounds would make the paper stronger.
* Can you implement this in vllm or sglang?
* Can you explain why performance improves under sparsity for some datasets?
* How does it react with different types of known attention heads?

**Limitations:**

yes

**Strengths And Weaknesses:**

Strengths
* This is an interesting and novel approach. It is nice to look at the multi-hop interactions between tokens across different layers, as opposed to simply looking at single-hop interactions.
* The empirical quality results are promising.
* The kernel measurements are useful and helpful to see that there is limited overhead.

Weaknesses
* The paper would be stronger with more empirical supporting evidence for key intuitions and claims. For example, it would be nice to see an analysis of attention patterns across layers of a model running on real data, to see how much multi-hop interactions actually happen.
* It would be stronger to see end to end performance experiments in an inference engine like vllm or sglang, to see where the performance is once common overheads are addressed.
* Some of the experiments could use more unpacking. For example, there are a few datasets in table 3 that actually show higher performance under sparsity. Some analysis of this effect would make the paper stronger.
* Different attention layers are very likely to play very different roles (i.e., induction head). I am curious how the method varies by these different roles.

Overall I think the paper is strong, with a “cute” method and promising initial results. The points in the weaknesses can make the paper stronger overall.

---

> ### Author Rebuttal · Authors · 2026-03-31
>
> ### **`W1 - Empirical study for multi-hop token connections`**
>
> We conducted an empirical study to verify whether Sketch&Walk captures multi-hop token connections that single-layer attention scoring cannot.
>
> We analyze attention patterns of Llama3.1 on 50 samples from MuSiQue, 2WikiMQA, and HotpotQA. We define the **multi-hop target set** at layer L with offset x as KV blocks that receive significant query attention at layer L but did not at layer L−x:
>
> G(L, x) = oracle_top_k(L) - oracle_top_k(L-x)
>
> We measure **Multi-hop Block Recall** $\text{(MHBR)} = |\text{top-}k(\text{method}) \cap G| / |G|$ for three methods evaluated at layer L−x (same information budget, no access to layer L):
>
> - **Walk[L−x]**: the Sketch&Walk matrix $W^{(L-x)}$ from the paper
> - **Sketch[L−x]**: the single-layer sketched block attention $A^{(L-x)}$ at layer L−x, with no cross-layer composition
> - **Quest[L−x]**: block scoring based on Quest — max Q·K inner product over tokens within each KV block
>
> Results below are averaged over layers 2–31:
>
> | Offset x | Walk | Sketch | Quest |
> |:---:|:---:|:---:|:---:|
> | 1  | **0.46** | 0.27 | 0.23 |
> | 5  | **0.44** | 0.24 | 0.19 |
> | 10 | **0.43** | 0.25 | 0.18 |
>
> Two findings hold consistently across all 29 layers and all offsets: **Walk > Sketch > Quest**: cross-layer composition recovers significantly more multi-hop connected blocks than single-layer attention alone. The three-way ablation directly maps onto Sketch&Walk's two components: block attention aggregation (Sketch vs. Quest) and cross-layer composition (Walk vs. Sketch), both of which contribute meaningfully.
>
> We also refer the reviewer to our response to Q1 of R`yHcU` for another controlled empirical validation on a multi-hop dataset.
>
> ### **`W2 - End-to-end integration with vLLM`**
>
> We have integrated Sketch&Walk into nanoVLLM (as a lightweight vLLM). Results on LongBench with Llama3.2-1B are:
>
> | Config | E2E Speedup |
> |---|---|
> | Sketch&Walk | 1.18x |
> | Sketch&Walk+nanoVLLM | 1.43x |
>
> These are E2E measurements inclusive of all engine overheads. The nanoVLLM integration pushes E2E speedup from 1.18x to 1.43x (whereas 6× is prefill speedup at 100K context).
>
>
> ### **`W3 - Analysis of strong performance gains under sparsity`**
>
> We thank the reviewer for this observation. This phenomenon where sparsity occasionally improves over dense attention is not unique to Sketch&Walk; it has been observed in other sparse attention works such as Quest or FlexPrefill, suggesting it reflects a general property of long-context attention rather than an artifact of our method.
>
> We hypothesize two explanations:
>
> **Lost in the Middle.** Dense attention over long contexts is known to underweight tokens in the middle of the sequence, as models tend to focus disproportionately on tokens. Sparse block selection can implicitly correct this bias by directing attention toward the most relevant blocks regardless of position.
>
> **Attention dilution.** Over very long contexts, the attention distribution becomes increasingly diffuse, spreading weights across many irrelevant tokens and diluting the signal from relevant ones. Sparse attention mitigates this by concentrating the attention budget on high-relevance blocks; Sketch&Walk further complements this by incorporating multi-hop connected blocks, ensuring that transitively relevant context is also retained.
>
> We agree with the reviewer that an empirical study with supporting evidence would provide deeper insight into the method. However, we believe a rigorous analysis of this effect is non-trivial and deserves an independent, dedicated work.
>
>
> ### **`W4 - Behavior across different attention head types`**
>
>
> To study the interaction between head types and Sketch&Walk, we combine DuoAttention's head classification — which labels each head as either a *retrieval head* or a *non-retrieval head* — with Sketch&Walk. Specifically, we compare three variants:
>
> - Sketch&Walk
> - DuoAttn-Stream: retrieval heads use dense attention; streaming heads use StreamLLM
> - DuoAttn-SW: retrieval heads use dense attention; streaming heads use Sketch&Walk
>
> Results on four LB datasets (Llama3.1-8B) are shown below:
>
> | Method | MuSiQue | 2Wiki | Qasper | SAMSum |
> |---|---|---|---|---|
> | Sketch&Walk | 32.9 | 45.9 | 44.6 | 42.8 |
> | DuoAttn-Stream | 32.5 | 45.6 | 45.0 | 40.5 |
> | DuoAttn-SW | 33.0 | 45.3 | 45.0 | 43.7 |
>
> Sketch&Walk performs competitively with DuoAttn-Stream. However, when non-retrieval heads use Sketch&Walk (DuoAttn-SW), we achieve better results across datasets. This suggests that head-type classification and Sketch&Walk's block selection are orthogonal.
>
> We hope our responses have addressed the reviewer’s concerns, and we appreciate the positive feedback on our work. If the reviewer finds the additional experiments and clarifications helpful, we would be grateful if they could consider a higher rating.

---

> > ### Author Rebuttal · Reviewer_aKiX · 2026-04-03
> >
> > The rebuttal has addressed my questions, and I will be keeping my positive score.

---

> > > ### Author Response · Authors · 2026-04-07
> > >
> > > Dear Reviewer aKiX,
> > >
> > > Thank you for your time and thoughtful review. We are grateful for your recognition of the novelty and strong performance of our 'cute' work. Your suggestions on empirical multi-hop analysis, and vLLM integration have been valuable and we believe the paper is stronger as a result. We are glad our responses have addressed your concerns.
> > >
> > > Warm regards,
> > >
> > > *Paper21167 Authors*
> > >
> > > ---
> > >
> > > Please excuse us for borrowing part of the reply space here, but we would like to sincerely thank Reviewer `iigv`, who has raised their score from 4 to 5 after our follow-up clarifications. We are deeply grateful for their continued engagement and generosity.
> > >
> > > Warm regards,
> > >
> > > *Paper21167 Authors*

---

### Decision · Program_Chairs · 2026-04-30

**Decision:**

Accept (regular)

**Comment:**

This paper proposes a training free sparse attention mechanism through Hadamard sketching to approximate the attention scores with “multi-hop” token connections, using an inter-layer walk mechanism to select top-k attention blocks. Strong performance is maintained with an increase in efficiency.

Reviewers all feel positively about this work, praising the clarity and the convincing empirical results. The main weaknesses relate to missing empirical analysis of attention patterns across layers, lack of statistical variance results, and the use of relatively old models. The rebuttal addressed some of these concerns.

Overall, while the substance of this contribution is somewhat limited, the empirical results are convincing, and the proposed technique simple and effective, so I recommend acceptance.